# Urotensin II-related peptides, Urp1 and Urp2, control zebrafish spine morphology

Elizabeth A Bearce, Zoe H Irons, Johnathan R O'Hara-Smith, Colin J Kuhns, Sophie I Fisher, William E Crow, Daniel T Grimes*

Institute of Molecular Biology, Department of Biology, University of Oregon, Eugene, United States

**Abstract** The spine provides structure and support to the body, yet how it develops its characteristic morphology as the organism grows is little understood. This is underscored by the commonality of conditions in which the spine curves abnormally such as scoliosis, kyphosis, and lordosis. Understanding the origin of these spinal curves has been challenging in part due to the lack of appropriate animal models. Recently, zebrafish have emerged as promising tools with which to understand the origin of spinal curves. Using zebrafish, we demonstrate that the urotensin II-related peptides (URPs), Urp1 and Urp2, are essential for maintaining spine morphology. Urp1 and Urp2 are 10-amino acid cyclic peptides expressed by neurons lining the central canal of the spinal cord. Upon combined genetic loss of Urp1 and Urp2, adolescent-onset planar curves manifested in the caudal region of the spine. Highly similar curves were caused by mutation of Uts2r3, an URP receptor. Quantitative comparisons revealed that urotensin-associated curves were distinct from other zebrafish spinal curve mutants in curve position and direction. Last, we found that the Reissner fiber, a proteinaceous thread that sits in the central canal and has been implicated in the control of spine morphology, breaks down prior to curve formation in mutants with perturbed cilia motility but was unaffected by loss of Uts2r3. This suggests a Reissner fiber-independent mechanism of curvature in urotensin-deficient mutants. Overall, our results show that Urp1 and Urp2 control zebrafish spine morphology and establish new animal models of spine deformity.

*For correspondence:
dtgrimes@uoregon.edu

**Competing interest:** The authors declare that no competing interests exist.

## Editor's evaluation

This is a beautifully executed study on the role of Urp signaling in spine morphogenesis in zebrafish. This work challenges the model that Urp1/ 2 controls the extension and straightening of the body axis of the zebrafish embryos, and thus makes a significant contribution to the literature.

## Introduction

Understanding how the shape of organisms is acquired is a central goal of developmental biology. The chordate body axis forms during embryonic development, when it is based around the rod-like notochord (*Stemple, 2005*). Later, the vertebrate axis comprises a column of repeating vertebrae which grows during juvenile and adolescent phases and is then maintained during adult life for up to several decades in some species (*Bagnat and Gray, 2020*). While a great deal has been learned about how the body axis emerges during embryogenesis, less is known about how spine morphology is maintained during growth and adulthood.

A breakdown of spine morphology occurs in scoliosis, lordosis, and kyphosis. Scoliosis is medically defined as lateral curvatures of the spine greater than 10° (*Cheng et al., 2015*; *Mesiti, 2021*; *Wise et al., 2008*) and can be caused by congenital defects of vertebral patterning or as a secondary consequence of neuromuscular disease (*Pourquié, 2011*; *Wishart and Kivlehan, 2021*). However,

**eLife digest** The backbone, or spine, is an integral part of the human body, providing support to our torsos so that we can sit, stand, bend and twist. If this structure does not form correctly, it can lead to pain, neurologic problems, and mobility issues. The spine normally has curves, but these can become deformed for many reasons, including genetic and muscular factors. There are also cases in which the cause of a spine distortion is unknown, such as in scoliosis (where the spine twists to the side), lordosis (where the lower part of the spine curves excessively), and kyphosis (where the upper part of the spine shows extreme curvature).

The structure of the spine is laid out during embryonic development and maintained throughout life. Experiments in zebrafish have shown that a crucial element in preserving the shape of the spine is the flow of cerebrospinal fluid or CSF. Propelled by the movement of little 'hairs' at the surface of specialized cells, this liquid runs through our central nervous system along a cavity lined with neurons. These nerve cells produce Urp1 and Urp2, two short molecules (or peptides) built from the same components as proteins. In zebrafish embryos, lowering the levels of these peptides had previously been shown to cause early body deformities. But what role, if any, do Urp1 and Urp2 play in maintaining the shape of the spine in adult zebrafish?

Bearce et al. set out to answer this question. First, they generated mutant zebrafish which did not carry either Urp1, Urp2 or both peptides. Contrary to previous findings, all three of these mutants developed normally as embryos. Once they were adults, zebrafish lacking Urp1 exhibited normal spines, while those lacking Urp2 had slightly deformed curves. However, zebrafish lacking both peptides had prominent curves in the tail-region of their spines, somewhat akin to lordosis in humans. This indicates that both peptides are necessary for adult spine structure, but work in a semi-redundant manner. Interestingly, the defects observed first appeared in adolescent fish and gradually worsened as they grew; many forms of human spinal abnormalities follow a similar trajectory.

Bearce et al. also tested the role of the protein Uts2r3, a receptor for peptides which belong to the urotensin family (such as Urp1 and Urp2). Fish lacking this protein showed normal spine structure as embryos, but distorted spinal curves as adults, suggesting that Urp1 and Urp2 might control spine morphology by signaling via the Uts2r3 receptor.

Together, Bearce et al.'s observations show that disturbing urotensin signaling leads to a lordosis-like condition in adult zebrafish, with evident deformities in the tail-region of the spine. Considering the broad similarities in structures between the zebrafish and the human spine, these results point to a possible involvement of urotensin signaling in spine distortion in humans. More studies using zebrafish will likely provide further insights into the principles that control the shape of the spine and what goes wrong when it breaks down.

most cases of scoliosis are idiopathic in nature, with no known etiology: approximately 3% of children are afflicted by idiopathic scoliosis, which most often onsets during adolescence (*Cheng et al., 2015*; *Labrom et al., 2021*). By contrast, kyphosis and lordosis occur when there is excessive curvature of the thoracic and lumbar regions of the vertebral column, respectively, resulting in a hunched upper back (kyphosis) or a concave lower back (lordosis) (*Ogura et al., 2021*) without vertebral structural defects. Since these categories of curves can co-occur, there are likely to be overlapping as well as distinct causes.

A challenge to understanding the origin of spinal curvature has been the dearth of suitable animal models recapitulating disease states. Recently, teleost fishes, in particular zebrafish (*Danio rerio*), have emerged as prominent animal models of spinal deformity (*Bagnat and Gray, 2020*; *Bearce and Grimes, 2021*; *Boswell and Ciruna, 2017*; *Gorman and Breden, 2009*; *Roy, 2021*). Using zebrafish, it was found that motile cilia-generated cerebrospinal fluid (CSF) flow is essential for maintaining body and spine morphology (*Grimes et al., 2016*). Mutants with defective motile cilia failed to undergo axial straightening during embryogenesis and so developed a misshapen early embryonic body axis called 'curly tail down' (CTD; *Brand et al., 1996*). If rescued during this early stage, mutants went on to develop three-dimensional spinal curves that recapitulated some features of idiopathic scoliosis, including adolescent-stage onset in the absence of vertebral patterning defects (*Grimes et al., 2016*; *Marie-Hardy et al., 2021*; *Wang et al., 2022*). Precisely how motile cilia and CSF flow maintain spine

morphology during growth is not understood, but it is known that during early larval stages cilia motility is essential for the assembly of the Reissner fiber (RF), an extracellular thread-like structure composed predominantly of the large glycoprotein SCOspondin (encoded by *sspo*) which sits in the CSF in brain ventricles and the central canal (*Cantaut-Belarif et al., 2018*; *Rodríguez et al., 1998*). Zebrafish *sspo* mutants exhibited CTD as embryos while hypomorphic mutants which can survive beyond embryonic stages also manifested spinal curves (*Cantaut-Belarif et al., 2018*; *Lu et al., 2020*; *Rose et al., 2020*; *Troutwine et al., 2020*).

The URPs, Urp1 and Urp2, may also function downstream of motile cilia in the central canal. Urp1 and Urp2 are 10-amino acid cyclic peptides previously linked to heart disease and mental illness (*Sugo et al., 2003*; *Konno et al., 2013*; *Nobata et al., 2011*; *Parmentier et al., 2011*; *Quan et al., 2021*; *Tostivint et al., 2006*; *Vaudry et al., 2010*). In zebrafish, Urp1 and Urp2 are expressed in CSF-contacting neurons (CSF-cNs), flow sensory neurons in the central canal, and their expression is increased by motile cilia function and the RF (*Cantaut-Belarif et al., 2020*; *Lu et al., 2020*; *Quan et al., 2015*; *Zhang et al., 2018*). Morpholino knockdown of Urp1/Urp2 results in embryonic CTD phenotypes while addition of Urp1/Urp2 peptides can rescue the CTD of cilia motility- and RF-deficient mutants (*Lu et al., 2020*; *Zhang et al., 2018*). This suggested that Urp1 and Urp2 act downstream of cilia motility to promote early axial straightening (*Grimes, 2019*; *Lu et al., 2020*; *Zhang et al., 2018*).

Here, we set out to address whether Urp1 and Urp2 function beyond embryogenesis in maintaining body and spine morphology during growth and adulthood. By generating zebrafish mutants lacking Urp1 and Urp2 peptides, we found that they are essential, in a semi-redundant fashion, for adult spine morphology. Loss of Urp1 and Urp2 together led to the onset of spinal curves during adolescent stages and, by adulthood, resulted in planar curves in the caudal region of the spine that occurred without vertebral patterning defects or significant structural malformations. A similar phenotype was present upon mutation of the urotensin receptor (UT) gene, *uts2r3*, suggesting that Urp1 and Urp2 signal via Uts2r3 to maintain spine morphology. Urotensin-associated curves were quantitatively distinct from the curves displayed by *cfap298* mutants, which lack cilia motility, and *pkd2l1* mutants in which a CSF-cN-localized ion channel is mutated, suggestive of overlapping but distinct roles of these components. Moreover, RF breakdown preceded curve formation in *cfap298* mutants while RF structure was maintained before and after curves appeared in *uts2r3* mutants. Overall, this demonstrates that Urp1 and Urp2 peptides control the morphology of the zebrafish spine. We suggest that urotensin-deficient zebrafish model human spinal deformities and will be important tools for deciphering how the spine is maintained and how this process goes wrong in disease.

## Results

### Urp1 and Urp2 peptides are dispensable for embryonic axial straightening

To determine whether Urp1 and Urp2 are required for spine morphology, we used CRISPR/Cas9 to generate zebrafish mutant lines. Urp1 and Urp2 are encoded by 5-exon genes with the final exon coding for the 10-amino acid peptides that are released by cleavage from the pro-domain (*Figure 1A–B*, *Figure 1—figure supplement 1A–B*). We used pairs of guide RNAs to induce deletions across the genetic region coding for the peptides (*Figure 1A*, *Figure 1—figure supplement 2A–B*). We refer to the resulting mutant lines as *urp1^ΔP^* and *urp2^ΔP^* because they lack the peptide coding sequence. In addition, mRNA quantitation revealed downregulation of *urp1* and *urp2* in their respective mutant backgrounds, indicating transcript decay (*Figure 1—figure supplement 2E*).

We first assessed *urp1^ΔP^* and *urp2^ΔP^* mutants for embryonic phenotypes. A previous morpholino-based knockdown study concluded that Urp1 and Urp2 are required for axial straightening, the process by which the ventrally curved zebrafish embryo straightens as the trunk elongates and detaches from the yolk (*Figure 1C*). Urp1/Urp2 morphants failed to undergo straightening and therefore displayed CTD (*Zhang et al., 2018*). Surprisingly, both *urp1^ΔP^* and *urp2^ΔP^* mutants underwent normal axial straightening and did not exhibit CTD (*Figure 1Ei*). By contrast, we observed CTD in both *cfap298^tm304^* mutants that lack cilia motility in the central canal (*Bearce et al., 2022*) and *sspo^b1446^* mutants in which the RF constituent SCOspondin is mutated, as expected (*Figure 1Ei*, *Figure 1—figure supplement 3A–B*). Notably, *cfap298^tm304^* and *sspo^b1446^* mutants maintained *urp1* and *urp2* expression in CSF-cNs,

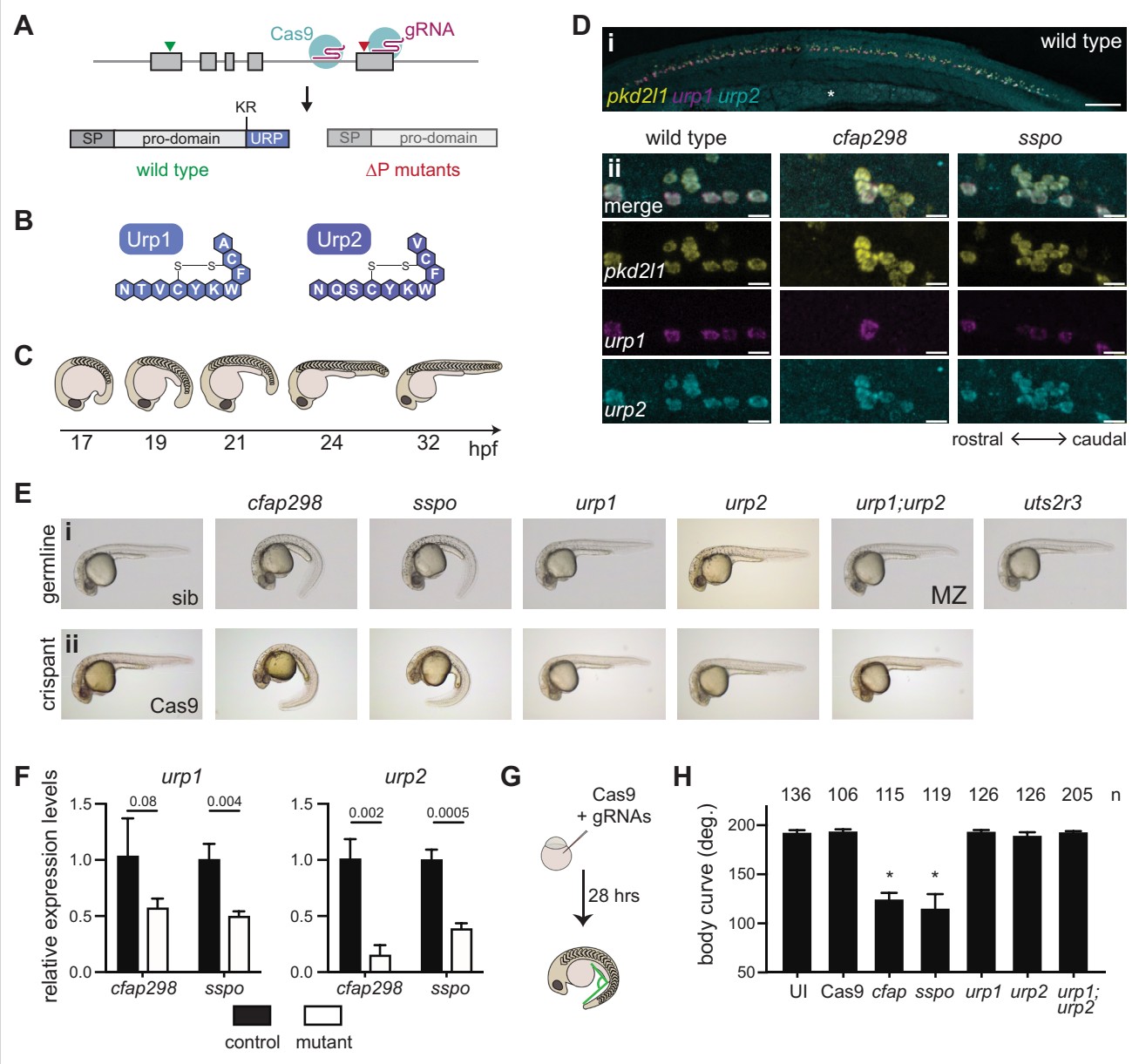

**Figure 1.** Urp1 and Urp2 are dispensable for axial straightening. (**A**) *urp1* and *urp2* are 5-exon genes (gray boxes). The final exon codes for the 10-amino acid peptides produced after cleavage from the prodomain at a dibasic site (KR). Pairs of gRNAs were used to induce deletions of Urp1 and Urp2 peptide coding sequences, resulting in *urp1^ΔP^* and *urp2^ΔP^* mutants, respectively. SP – signal peptide. (**B**) Urp1 and Urp2 peptide sequences with identical hexacyclic regions. (**C**) Zebrafish posterior axial straightening, the morphogenetic process which straightens the embryonic body. (**D**) Fluorescence in situ hybridization based on hybridization chain reaction analysis of *pkd2l1*, *urp1*, and *urp2* expression in the central canal at 28 hpf. *pkd2l1* expression marks CSF-cNs. *urp1* expression is restricted to ventral CSF-cNs while *urp2* is expressed in all CSF-cNs. Both *urp1* and *urp2* are expressed in *cfap298^tm304^* and *sspo^b1446^* mutants, though comparison of expression between samples was non-quantitative. (**i**) Shows the zebrafish trunk with the yolk stalk labeled (*). (**ii**) Shows zoomed regions taken at the rostro-caudal level at the end of the yolk stalk. Scale bars: 150 µm (**i**), 10 µm (**ii**). (**E**) Lateral views of 28–30 hpf germline mutants (**i**) and crispants (**ii**). The *urp1^ΔP^;urp2^ΔP^* double mutants are maternal zygotic (MZ) mutants. Sibling (sib) and Cas9-only injected embryos served as controls. All embryos were incubated at 28°C, which is a restrictive temperature for *cfap298^tm304^*. (**F**) Quantitative reverse transcriptase PCR (qRT-PCR) analysis of *urp1* and *urp2* mRNA expression levels in *cfap298^tm304^* and *sspo^b1446^* mutants at 28 hpf. n>3 biologically independent samples. Bars represent mean ± s.e.m. Two-tailed student's *t* test used to calculate p-values. (**G**) Schematic of crispant generation and body curve analysis. (**H**) Quantitation of crispant body curves where bars represent mean ± s.d. for at least three independent clutches and injection mixes. The total number of embryos analyzed is given. *p<0.0001, student's *t* test applied. UI – uninjected.

The online version of this article includes the following source data and figure supplement(s) for figure 1:

**Source data 1.** Raw data for qRT-PCR and crispant body angle measurements.

*Figure 1 continued on next page*

*Figure 1 continued*

**Figure supplement 1.** Urotensin family peptides.

**Figure supplement 2.** Generation of *urp1^{ΔP}* and *urp2^{ΔP}* mutants.

**Figure supplement 3.** Generation of *sspo^{b1446}* mutants.

central canal neurons marked by *pkd2l1* expression (*Figure 1D*). However, *urp1* and *urp2* transcripts were quantitatively reduced in *cfap298^{tm304}* and *sspo^{b1446}* mutants (*Figure 1F*). We reasoned that the absence of CTD in *urp1^{ΔP}* and *urp2^{ΔP}* mutants might reflect redundancy, since Urp1 and Urp2 peptides are highly similar, with identical hexacyclic regions (*Figure 1B*, *Figure 1—figure supplement 1A–B*). Alternatively, maternally derived *urp1* and/or *urp2* transcripts may function to prevent phenotypes from manifesting. However, maternal zygotic *urp1^{ΔP}*;*urp2^{ΔP}* double mutants also exhibited linear body axes (*Figure 1Ei*), ruling out redundant or maternal gene product function. This demonstrates that Urp1 and Urp2 peptide-null mutants undergo axial straightening.

To confirm this finding, we performed additional Urp1 and Urp2 loss-of-function experiments. By injecting four guide RNAs (gRNAs) along with Cas9 into wild-type embryos at the one-cell stage, we generated mosaic mutants, called crispants, that were then assessed for body shape phenotypes (*Figure 1G*). In positive control experiments, *cfap298* and *sspo* crispants exhibited robust CTD, phenocopying germline *cfap298^{tm304}* and *sspo^{b1446}* mutants (*Figure 1Eii*). Quantitation of body curvature revealed that crispant generation was highly efficient, with CTD penetrance being close to 100% (*Figure 1H*). By contrast, *urp1* and *urp2* single and double crispants exhibited straight body axes that were not different to uninjected embryos or embryos injected with Cas9 only (*Figure 1Eii and H*). Using T7 endonuclease assays, we confirmed that high levels of insertion-deletion mutations were generated at gRNA sites in crispants (*Figure 1—figure supplement 2C–D*). We used the AB genetic background for the majority of our work, but we also generated and phenotyped *urp1*;*urp2* double crispants on WIK and TU backgrounds to test for potential background effects. Normal axial straightening upon mutation of *urp1* and *urp2* was also observed on these backgrounds (*Figure 1—figure supplement 2F*). Overall, crispant results confirmed germline mutant findings. We conclude that Urp1 and Urp2 peptides are dispensable for axial straightening in embryonic zebrafish.

## Urp1 and Urp2 function semi-redundantly to maintain spine morphology

Next, we determined the impact of Urp1 and Urp2 loss on adult spine morphology. Outwardly, *urp1^{ΔP}* mutant adults at 3 months post fertilization (mpf) appeared normal whereas *urp2^{ΔP}* mutants exhibited minor body dysmorphologies and kinked tails (n=72 for *urp1^{ΔP}* mutants and n=92 for *urp1^{ΔP}* mutants, *Figure 2—figure supplement 1A*). To assess spine morphology directly, we imaged bone by X-ray microcomputed tomography (μCT). Three-dimensional reconstitutions of μCT data from 3 mpf fish showed that *urp1^{ΔP}* mutants indeed exhibited overtly normal skeletal morphology (n=7) while *urp2^{ΔP}* mutants showed slight sagittal curves (n=4; *Figure 2—figure supplement 2A*, *Figure 2—videos 1–3*). By contrast to these absent or mild deformities in single mutants, *urp1^{ΔP}*;*urp2^{ΔP}* double mutants exhibited prominent curves, with significant dorsal-ventral Cobb angles, a measure of deviation from straightness, especially in the caudal region of the spine (*Figure 2B and D–F*, *Figure 2—figure supplements 1A and 2A*, *Figure 2—videos 1–4*). These data indicate that Urp1 and Urp2 are essential for adult spine morphology, and that they function in a semi-redundant fashion in this context.

To assess the long-term maintenance of spine morphology in Urp1- and Urp2-deficient conditions, we aged *urp1^{ΔP}* and *urp2^{ΔP}* single mutants to 12 mpf then performed μCT. At this later time point, *urp1^{ΔP}* and *urp2^{ΔP}* mutants exhibited mild kyphosis-like curves though *urp2^{ΔP}* mutants were more severe (*Figure 2—figure supplement 2B*, *Figure 2—videos 5–7*). These degenerative phenotypes demonstrate that Urp1 and Urp2 are essential for maintenance of spine morphology throughout adulthood and aging, and suggest that Urp2 plays a larger role than Urp1.

## Urp1 and Urp2 signal through the Uts2r3 receptor to control spine morphology

Urp1 and Urp2 peptides engage G-protein-coupled receptors (*Ames et al., 1999*; *Chatenet et al., 2004*; *Elshourbagy et al., 2002*; *Labarrère et al., 2003*; *Liu et al., 1999*; *Nothacker et al., 1999*).

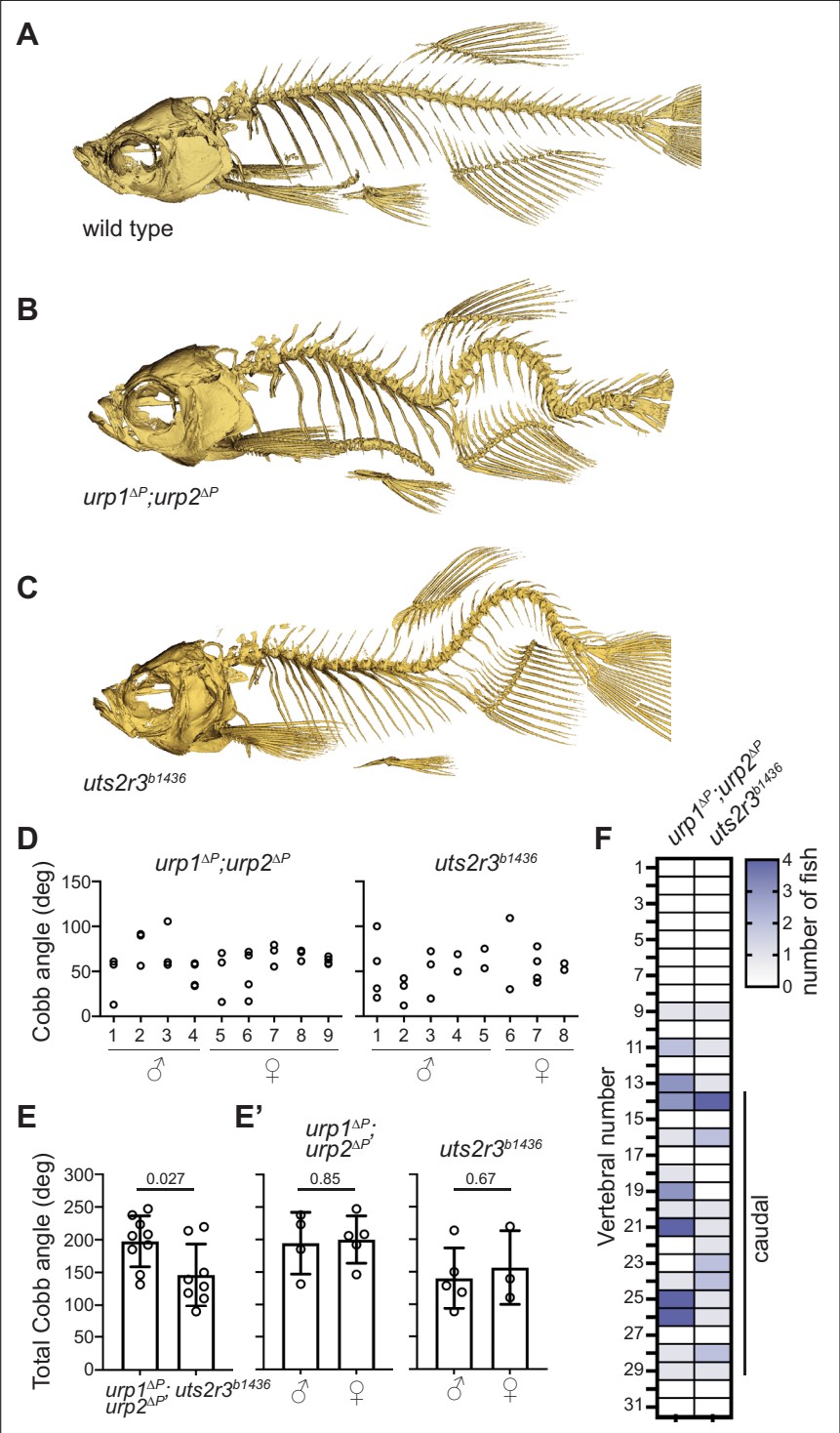

**Figure 2.** Urp1 and Urp2 are required for proper adult spine morphology. (**A–C**) Lateral views of microcomputed tomography reconstitutions of wild-type (**A**), *urp1^ΔP^;urp2^ΔP^* (**B**) and *uts2r3^b1436^* (**C**) mutants at 3 mpf. (**D**) Cobb angle measurements for individual fish in the sagittal plane for *urp1^ΔP^;urp2^ΔP^* and *uts2r3^b1436^* mutants. Circles represent angles for individual curves. (**E-E'**) Total Cobb angles with each circle representing an individual fish. The mean ± s.d. is shown. (**G'**) is the data from G parsed for sex. p-Values are given from two-tailed unpaired student's *t* tests. (**F**) The position of curve apex is plotted and shows that most curves are in caudal vertebrae. n=9 and 8 for *urp1^ΔP^;urp2^ΔP^* and *uts2r3^b1436^* mutants, respectively.

The online version of this article includes the following video, source data, and figure supplement(s) for figure 2:

*Figure 2 continued on next page*

*Figure 2 continued*

**Source data 1.** Raw data from spinal curve phenotypic measurements.

**Figure supplement 1.** Phenotyping spinal curves.

**Figure supplement 1—source data 1.** Raw data from spinal curve phenotypic measurements.

**Figure supplement 2.** Spinal curves in *urp1^ΔP^*, *urp2^ΔP^*, *urp1^ΔP^;urp2^ΔP^*, and *pkd2l1^icm02^* mutants degenerate with age.

**Figure supplement 3.** Generation of *uts2r3^b1436^* mutants.

**Figure 2—video 1.** Three-dimensional reconstitution of a 3-mpf wild-type male zebrafish. https://elifesciences.org/articles/83883/figures#fig2video1

**Figure 2—video 2.** Three-dimensional reconstitution of a 3-mpf *urp1^ΔP^* male zebrafish. https://elifesciences.org/articles/83883/figures#fig2video2

**Figure 2—video 3.** Three-dimensional reconstitution of a 3-mpf *urp2^ΔP^* male zebrafish. https://elifesciences.org/articles/83883/figures#fig2video3

**Figure 2—video 4.** Three-dimensional reconstitution of a 3-mpf *urp1^ΔP^;urp2^ΔP^* male zebrafish. https://elifesciences.org/articles/83883/figures#fig2video4

**Figure 2—video 5.** Three-dimensional reconstitution of a 12-mpf wild-type male zebrafish. https://elifesciences.org/articles/83883/figures#fig2video5

**Figure 2—video 6.** Three-dimensional reconstitution of a 12-mpf *urp1^ΔP^* male zebrafish. https://elifesciences.org/articles/83883/figures#fig2video6

**Figure 2—video 7.** Three-dimensional reconstitution of a 12-mpf *urp2^ΔP^* male zebrafish. https://elifesciences.org/articles/83883/figures#fig2video7

**Figure 2—video 8.** Three-dimensional reconstitution of a 3-mpf *uts2r3^b1436^* male zebrafish. https://elifesciences.org/articles/83883/figures#fig2video8

While a single urotensin II receptor (UT) gene is found in humans and has recently been linked to abnormal spinal curvature (*Dai et al., 2021*), the zebrafish genome encodes five such receptors. One of those, Uts2r3, was previously implicated in spine morphology (*Zhang et al., 2018*). To systematically compare the effects of Uts2r3 receptor mutation with loss of Urp1 and Urp2, we generated a *uts2r3* mutant line harboring a 178-amino acid deletion after the third amino acid, significantly disrupting the protein (*Figure 2—figure supplement 3*). Like *urp1^ΔP^;urp2^ΔP^* double mutants, these *uts2r3^b1436^* mutants underwent normal axial straightening as embryos (*Figure 1Ei*) and went on to exhibit spinal curves as adults (*Figure 2C*, *Figure 2—figure supplement 1A*, *Figure 2—video 8*). Cobb angle measurements showed that *uts2r3^b1436^* mutants and *urp1^ΔP^;urp2^ΔP^* mutants were similar, though curves in *urp1^ΔP^;urp2^ΔP^* mutants were slightly more severe (*Figure 2D–E*). Like *urp1^ΔP^;urp2^ΔP^* mutants, *uts2r3^b1436^* mutants showed mostly caudally located curves, especially in the most rostral of the caudal vertebrae (*Figure 2F*). Thus, although we cannot rule out minor roles for other UT receptors, these data suggest that Urp1 and Urp2 control spine morphology largely by signaling through Uts2r3.

## Urotensin pathway mutants display adolescent-onset spinal curves in the absence of structural vertebral defects

Next, we determined whether urotensin pathway mutants recapitulated any signs of disease present in patients. Several types of human spinal curves onset during adolescent growth (*Cheng et al., 2015*). To discern the stage of onset of curves in urotensin pathway mutants, we monitored *urp1^ΔP^;urp2^ΔP^* double mutant cohorts as they grew. Subtle curves first became apparent between 9 and 11 days pf (dpf), corresponding to a standard length between 3.9±0.7 mm and 5.9±0.4 mm (mean ± s.d.; *Figure 3A–B*, *Figure 3—video 1*), a stage when adolescents were rapidly growing (*Figure 3C*). By 13 dpf (standard length 6.2±0.3 mm), curves were evident in all *urp1^ΔP^;urp2^ΔP^* mutants and progressively worsened up to 17 dpf (8.3±0.4 mm) when we ended this analysis (*Figure 3A–B*). At 1 mpf, we assessed *urp1^ΔP^;urp2^ΔP^* mutants by μCT and found variability in curve position and amplitude (*Figure 3D*). Notably, at this stage, five of seven mutants exhibited a significant curve in the pre-caudal vertebrae, in addition to a caudal curve (*Figure 3D*, *Figure 3—figure supplement 1Ai-Bii*). Since pre-caudal curves were rare in mutants at 3 mpf (*Figure 2B and F*), this suggested that curve

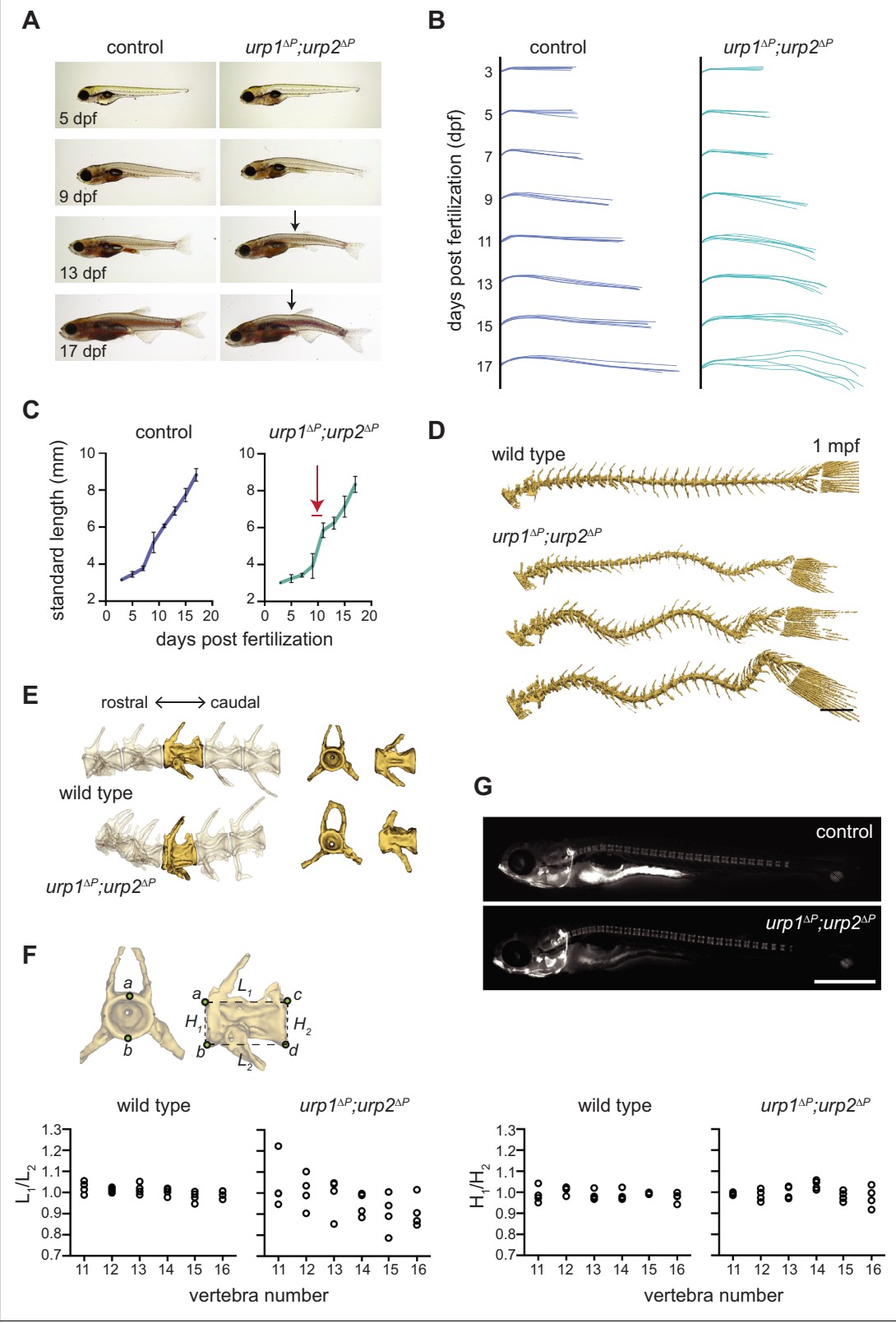

**Figure 3.** *urp1^(ΔP)*;*urp2^(ΔP)* mutants exhibit adolescent-onset spinal curves without significant structural vertebral defects. (**A**) Lateral views of control fish and age-matched *urp1^(ΔP)*;*urp2^(ΔP)* mutants. Arrows point to forming body curves. (**B**) Traces of body shape every 2 days for 5 fish per time point from 3 to 17 dpf. (**C**) Growth curves for control and *urp1^(ΔP)*;*urp2^(ΔP)* mutants were indistinguishable. Arrow shows time of curve onset. Mean ± s.d. is plotted. (**D**) Microcomputed tomography (μCT) reconstitutions of spines at 1 mpf with heads, fins, and ribs digitally removed. Scale bar: 1 mm. (**E**) μCT

*Figure 3 continued on next page*

Figure 3 continued

reconstitutions of three pre-caudal and two caudal vertebrae including frontal and lateral views of the highlighted vertebra. No major structural defects such as fusions were observed in $urp1^{\Delta P};urp2^{\Delta P}$ mutants. (F) Vertebral body rostral-caudal length ($L_1/L_2$) and dorsal-ventral height ($H_1/H_2$) aspect ratios for six vertebrae of wild type (n=4) and $urp1^{\Delta P};urp2^{\Delta P}$ mutants (n=4). Length aspect ratios were significantly more variable in mutants, but height aspect ratios were unchanged (p=0.022 and 0.745, respectively, Bartlett's test for equal variances). (G) Calcein staining revealed well-structured vertebrae forming in control (standard length 5.7 mm) and $urp1^{\Delta P};urp2^{\Delta P}$ mutant (standard length 5.7 mm) fish. n>30 fish per condition.

The online version of this article includes the following video, source data, and figure supplement(s) for figure 3:

**Source data 1.** Raw data from larval growth measurements and vertebral quantitation.

**Figure supplement 1.** Spinal curves are variable in 1 mpf $urp1^{\Delta P};urp2^{\Delta P}$ mutants.

**Figure 3—video 1.** Time course of juvenile development in wild-type and $urp1^{\Delta P};urp2^{\Delta P}$ siblings.
https://elifesciences.org/articles/83883/figures#fig3video1

**Figure 3—video 2.** Vertebral reconstitutions from wild-type, $urp1^{\Delta P};urp2^{\Delta P}$ and $uts2r3^{b1436}$ adults.
https://elifesciences.org/articles/83883/figures#fig3video2

location is dynamic and that pre-caudal curves form then resolve or shift in some mutants as they grow to adulthood.

Next, we assessed whether spinal curves in urotensin pathway mutants were caused by congenital defects of vertebral patterning or structure. Staining of juveniles with the vital dye calcein revealed no defects in vertebral patterning or spacing in $urp1^{\Delta P};urp2^{\Delta P}$ mutants at 10 dpf (4.5–6 mm standard length [*Figure 3C*]) (*Figure 3G*). We then assessed µCT data from 3 mpf fish and quantified vertebral body shape for vertebrae 11–16, where curves occurred in $urp1^{\Delta P};urp2^{\Delta P}$ mutants (*Figure 2F*). Vertebral body length and height aspect ratios were 1.00±0.007 (mean ± s.d.) and 0.99±0.007, respectively, for wild type (n=4; *Figure 3F*). By contrast, $urp1^{\Delta P};urp2^{\Delta P}$ mutants exhibited more variable vertebral length aspect ratios (0.97±0.03, p=0.022; Bartlett's test for equal variances, *Figure 3F*). Vertebral height aspect ratios in mutants (1.00±0.006) were not significantly different to controls (p=0.745, *Figure 3F*). These data are consistent with subtle vertebral shape defects at the points of curvature in $urp1^{\Delta P}urp2^{\Delta P}$ mutants. However, we do not observe vertebral fusions or missing or transformed appendages, suggesting that vertebral defects do not underlie spinal curves; instead, small changes in vertebral shape are likely due to the presence of curves themselves rather than causative of curves in the first place. This is similar to what occurs in non-congenital forms of human scoliosis (*Cheng et al., 2015*).

Additionally, we parsed our phenotypic data for sex since spinal curves often show sex bias in severity in humans (*Cheng et al., 2015*), something which has been recapitulated in some zebrafish spinal curve mutants (*Marie-Hardy et al., 2021*). However, in both $urp1^{\Delta P};urp2^{\Delta P}$ and $uts2r3^{b1436}$ mutants, we found no significant differences in curve penetrance or severity between males and females (*Figure 2E'*).

## Urotensin-deficient mutants are phenotypically distinct from $cfap298^{tm304}$ and $pkd2l1^{icm02}$ mutants

We next compared the phenotypes of urotensin pathway mutants to other mutant lines that exhibit spinal curves. The $cfap298^{tm304}$ line harbors a temperature-sensitive mutation in $cfap298$, a gene required for cilia motility in several organisms including humans (*Austin-Tse et al., 2013*; *Bearce et al., 2022*; *Jaffe et al., 2016*). $cfap298^{tm304}$ mutants exhibit reduced cilia motility in the central canal and, if the resulting CTD is embryonically rescued by temperature shifts, develop adolescent-onset spinal curves (*Grimes et al., 2016*). These curves were argued to model an adolescent idiopathic scoliosis (AIS)-like condition (*Figure 4A–B*, *Figure 4—video 1*; *Grimes et al., 2016*; *Marie-Hardy et al., 2021*). Importantly, both $urp1$ and $urp2$ transcripts were significantly downregulated in $cfap298^{tm304}$ mutants (*Figure 1F*) suggesting that spinal curves in $cfap298^{tm304}$ might be the result of reduced Urp1/Urp2 expression. To systematically compare $cfap298^{tm304}$ mutants with urotensin-deficient mutants, we raised cohorts of $urp1^{\Delta P};urp2^{\Delta P}$ mutants, $uts2r3^{b1436}$ mutants, and temperature-shift-rescued $cfap298^{tm304}$ mutants alongside one another in the same aquatics facility after backcrossing all lines to the AB strain for multiple generations. At 3 mpf, we performed µCT scanning and three-dimensional reconstitutions. First, we calculated dorso-ventral Cobb angles, which revealed that $cfap298^{tm304}$ mutants were more severely curved (average total Cobb angle: 260.4±32.7°) than either $urp1^{\Delta P};urp2^{\Delta P}$ mutants (197.5±38.9°)

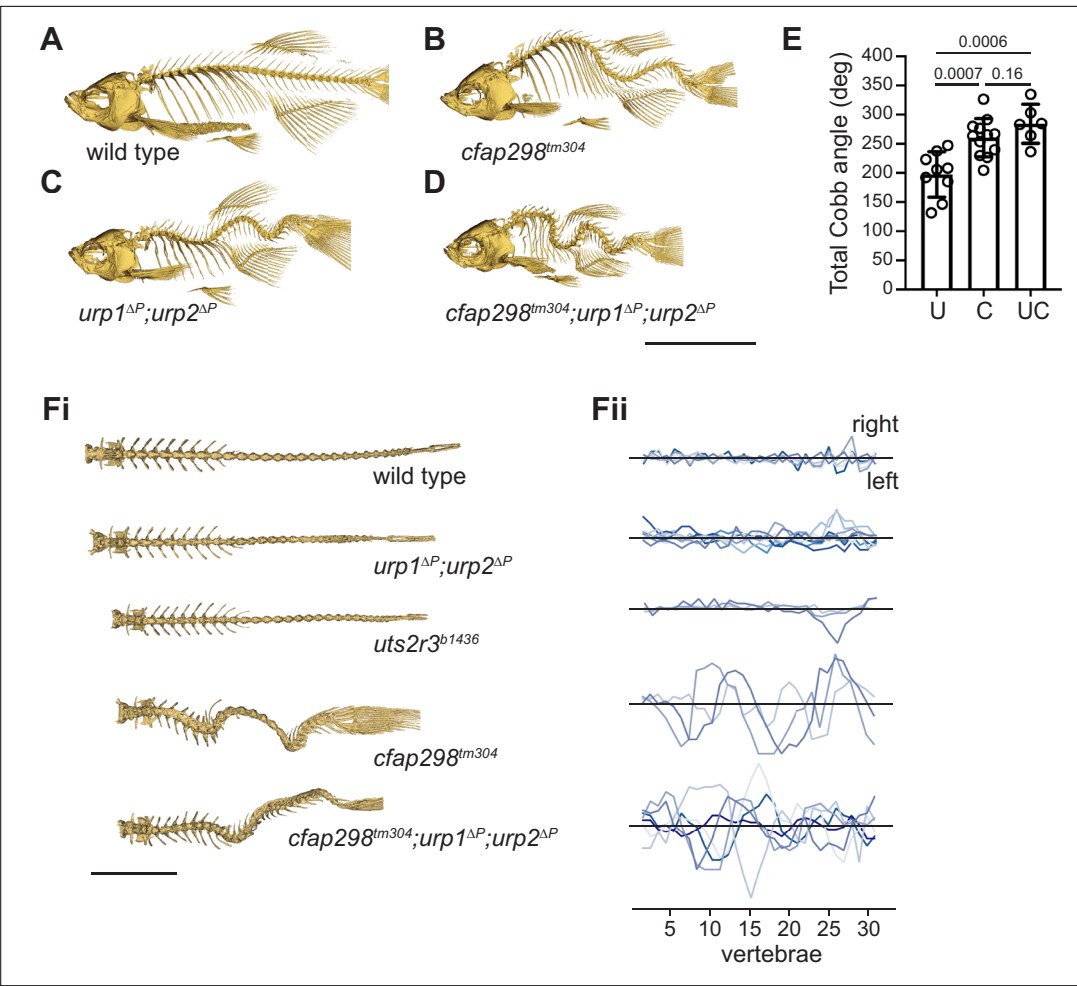

**Figure 4.** *urp1^ΔP;urp2^ΔP* mutants and *cfap298^tm304* mutants are phenotypically distinct. (**A–D**) Lateral views of microcomputed tomography (μCT) reconstitutions of wild-type (**A**), *cfap298^tm304* mutants (**B**), *urp1^ΔP;urp2^ΔP* double mutants (**C**), and *cfap298^tm304;urp1^ΔP;urp2^ΔP* triple mutants (**D**). All fish shown are female. Scale bar: 10 mm. (**E**) Total Cobb angles with each circle representing an individual fish. The mean ± s.d. is shown. p-Values are given from two-tailed unpaired student's *t* tests. U — *urp1^ΔP;urp2^ΔP* double mutants; C — *cfap298^tm304* mutants; UC — *cfap298^tm304;urp1^ΔP;urp2^ΔP* triple mutants. (**Fi**) Dorsal views of μCT reconstitutions with ribs and fins removed. Scale bar: 5 mm. (**Fii**) Quantitation of degree of lateral curvature for wild type (n=5) and *urp1^ΔP;urp2^ΔP* (n=8), *uts2r3^b1436* (n=3), *cfap298^tm304* (n=3), and *cfap298^tm304;urp1^ΔP urp2^ΔP* (n=6) mutants. y-axis is the arbitrary units.

The online version of this article includes the following video, source data, and figure supplement(s) for figure 4:

**Source data 1.** Raw data for quantion of spinal phenotypes.

**Figure supplement 1.** Analysis of lateral curvature.

**Figure 4—video 1.** Three-dimensional reconstitution of a 3-mpf *cfap298^tm304* male zebrafish.
https://elifesciences.org/articles/83883/figures#fig4video1

**Figure 4—video 2.** Three-dimensional reconstitution of a 3-mpf *cfap298^tm304;urp1^ΔP;urp2^ΔP* male zebrafish.
https://elifesciences.org/articles/83883/figures#fig4video2

**Figure 4—video 3.** Three-dimensional reconstitution of a 3-mpf *pkd2l1^icm02* male zebrafish.
https://elifesciences.org/articles/83883/figures#fig4video3

**Figure 4—video 4.** Three-dimensional reconstitution of a 12-mpf *pkd2l1^icm02* male zebrafish.
https://elifesciences.org/articles/83883/figures#fig4video4

or *uts2r3^b1436* mutants (146.1±47.4°) (*Figure 4E*, *Figure 2E*, *Figure 2—figure supplement 1D–E*). Second, *cfap298^tm304* mutants showed prominent dorsal-ventral curves in the pre-caudal as well as caudal vertebrae, a distinct pattern compared with the predominantly caudal curves in *urp1^ΔP;urp2^ΔP* and *uts2r3^b1436* mutants (*Figure 4A–C*, *Figure 2—figure supplement 1C*). Third, *cfap298^tm304* mutants exhibited significant lateral curvature of the spine, often with spinal twisting, a hallmark of AIS-like curves (*Figure 4Fi–Fii*, *Figure 4—figure supplement 1*). By contrast, *urp1^ΔP;urp2^ΔP* and *uts2r3^b1436* mutants showed planar curves, with very minor or no lateral deviations (*Figure 4Fi–Fii*, *Figure 4—figure supplement 1*). These results demonstrated that cilia motility mutants and urotensin-deficient mutants exhibit distinct spinal curve phenotypes. As such, the causes of spinal curves in *cfap298^tm304* mutants can only be partially explained by reduced Urp1/Urp2 expression.

To further explore the relationship between cilia motility and urotensin peptides, we generated adult *cfap298^tm304;urp1^ΔP;urp2^ΔP* triple mutants that were embryonically rescued by temperature shifts. Triple mutants exhibited significant curves, similar to *cfap298^tm304* single mutants (*Figure 4D–E*, *Figure 2—figure supplement 1C–E*, *Figure 4—video 2*; average total Cobb angle: 284.4±33.5°). The curves of triple mutants were three-dimensional in nature, with both dorso-ventral and medio-lateral deviations, as well as incidences of spinal torsion (*Figure 4Fi–Fii*, *Figure 4—figure supplement 1*); curves were present in both pre-caudal and caudal vertebrae, as opposed to being more restricted to caudal vertebrae in *urp1^ΔP;urp2^ΔP* double mutants (*Figure 2—figure supplement 1C*). Overall, this suggests that motile cilia contribute to urotensin-dependent and urotensin-independent pathways controlling spine morphology.

Pkd2l1 is a polycystin family ion channel expressed in CSF-cNs, the same cell type which expresses Urp1 and Urp2 (*Figure 1D*; *Quan et al., 2015*). Pkd2l1 is responsible for flow-induced Ca^2+ signaling in CSF-cNs (*Böhm et al., 2016*; *Sternberg et al., 2018*). While *pkd2l1^icm02* mutants exhibited normal early axis development, they went on to develop mild kyphosis-like curves upon aging (*Sternberg et al., 2018*), a result we recapitulated after raising *pkd2l1^icm02* mutants on the same genetic background and under the same conditions as urotensin-deficient mutants for direct comparison (*Figure 2—figure supplement 2A–B*). Notably, *pkd2l1^icm02* mutants showed very subtle curves in the pre-caudal vertebrae and absence of lateral deviation at 12 mpf but no obvious curves of any kind at 3 mpf (*Figure 2—figure supplement 2A-B*, *Figure 4—videos 3–4*). This mild kyphosis-like phenotype was therefore also highly distinct from *urp1^ΔP;urp2^ΔP* and *uts2r3^b1436* mutants.

Overall, our phenotypic data show that *urp1^ΔP;urp2^ΔP* mutants develop adolescent-onset curves without significant vertebral structural defects or sex bias. Coupled to the consistently caudal location of curves at 3 mpf as well as the lack of lateral deviation, we suggest that urotensin pathway mutants

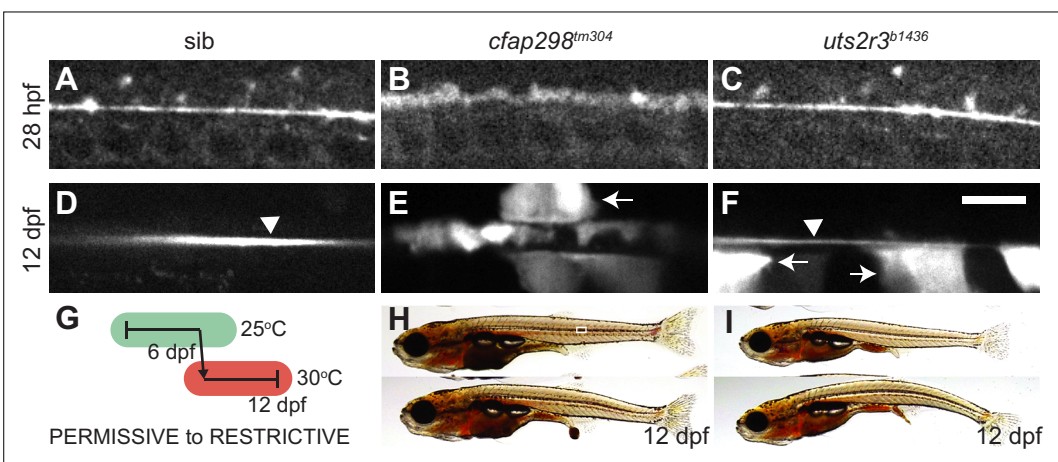

**Figure 5.** Reissner fiber (RF) breakdown in *cfap298^tm304* mutants but not urotensin-deficient mutants. (**A–F**) Grayscale maximal intensity projection of Sspo-GFP localization in the central canal in 28 hpf embryos (**A–C**) and 12 dpf adolescents (**D–F**). RF is denoted by arrow heads in D and F. Arrows point to structures along the central canal that become GFP-positive in *cfap298^tm304* and *uts2r3^b1436* mutants. Scale bar: 10 µm. (**G**) Schematic of temperature shift experiment in which *cfap298^tm304* mutants are initially raised at permissive temperatures before being shifted to restrictive temperatures at 6 dpf, then imaged at 12 dpf. (**H–I**) Lateral views of *cfap298^tm304* (**H**) and *uts2r3^b1436* (**I**) mutants at 12 dpf when Sspo-GFP imaging took place. The white box in H shows the location imaged in **D–F**.

most closely reflect a lordosis-like condition. In agreement, urotensin mutants were phenotypically distinct from established AIS-like (*cfap298^tm304*) and kyphosis-like (*pkd2l1^icm02*) models.

## RF breakdown precedes AIS-like curves in *cfap298^tm304* mutants

Given the links between motile cilia, the RF and *urp1* and *urp2* expression (*Figure 1D and F*; *Cantaut-Belarif et al., 2020*; *Lu et al., 2020*; *Zhang et al., 2018*) as well as the requirement for proper RF assembly to prevent spinal curves (*Rose et al., 2020*; *Troutwine et al., 2020*), we assessed Sspo, the major component of the RF, in spinal curve mutants. To visualize Sspo localization, we used the *sspo-GFP^ut24* line in which GFP coding sequence is fused to the endogenous *sspo* locus, producing Sspo-GFP protein (*Troutwine et al., 2020*). First, we assessed Sspo localization in the central canal of *cfap298^tm304* mutants at 28 hr pf (hpf). In sibling controls, Sspo localized into an RF throughout the central canal (*Figure 5A*). By contrast, *cfap298^tm304* mutants raised at restrictive temperatures, which exhibited reduced central canal cilia motility and CTD (*Figure 1E*; *Bearce et al., 2022*), lacked RF. Instead, Sspo was diffusely localized in the central canal in these mutants (*Figure 5B*), in agreement with previous work showing that cilia motility is required for RF assembly in embryos (*Cantaut-Belarif et al., 2018*).

Next, we took advantage of the temperature-sensitive nature of the *cfap298^tm304* mutation to determine whether RF is also disrupted at later timepoints, during adolescent stages when true spinal curves begin to develop. To do so, we initially raised *cfap298^tm304* mutants at permissive temperatures, allowing RF to correctly form and embryos to fully straighten. At 6 dpf, we transitioned larvae to restrictive temperatures (*Figure 5G*), which led to the appearance of curves by 11–14 dpf (standard length 6.5±0.2 mm), then assessed Sspo localization at 12 dpf. As in embryos, Sspo localized into a defined RF in the central canal in sibling controls (arrow head in *Figure 5D*) but was diffuse in temperature upshifted *cfap298^tm304* mutants, both in mutants that had yet to develop curves and those exhibiting subtle curves (*Figure 5E*, *Figure 5H*). This demonstrates (1) that cilia motility is not only required for the initial formation of the RF but also for its maintenance; and (2) breakdown of the RF precedes curve onset in cilia motility-deficient mutants. This supports a model in which continued cilia motility maintains the RF structure, and loss of the RF causes AIS-like curves to develop in cilia motility mutants.

## The RF remains intact in Uts2r3-deficient mutants both before and after curve formation

Next, we imaged Sspo-GFP localization in the central canal of *uts2r3^b1436* mutants to determine whether RF breakdown could be causative in urotensin-associated spinal curves. In agreement with that lack of axial straightening phenotype in *uts2r3^b1436* mutant embryos (*Figure 1Ei*), we found that Sspo localized normally into the RF in the central canal at 28 hpf in *uts2r3^b1436* mutants (*Figure 5C*). At 12 dpf, as spinal curves were beginning to manifest (*Figure 5I*), Sspo still formed an intact RF in *uts2r3^b1436* mutants (arrow head in *Figure 5F*). The fact that RF is present in *uts2r3^b1436* mutants as curves form suggests that urotensin-associated curves are not caused by defective RF formation. This result coheres with the distinct spinal phenotypes exhibited by *cfap298^tm304* mutants and *uts2r3^b1436* mutants.

While imaging Sspo-GFP at 12 dpf, we noted that in both *cfap298^tm304* mutants and *uts2r3^b1436* mutants, large central canal cells with the appearance of CSF-cNs became GFP-positive (arrows in *Figure 5E–F*), something we rarely observed in control fish (*Figure 5D*). Indeed, the GFP-positive central canal cells in *uts2r3^b1436* give the effect of making the RF appear comparatively smaller/dimmer (compare RF in *Figure 5F* and *Figure 5D*). We suggest that CSF-cNs may endocytose Sspo-GFP monomers in *cfap298^tm304* mutants where the RF has broken down and in *uts2r3^b1436* mutants where RF is likely to be making increasing numbers of contacts with CSF-cNs (*Orts-Del'Immagine et al., 2020*) owing to the onset of spinal curves.

## Discussion

Urotensin II (UII) is a cyclic peptide that was first identified from the teleost urophysis (*Pearson et al., 1980*) and subsequently found to exist in amphibians (*Conlon et al., 1992*) and mammals (*Coulouarn et al., 1998*). A highly similar peptide, called URP was then isolated from the brains of rodents (*Sugo*

*et al., 2003*). UII and URP both signal via the UT, a G-protein-coupled receptor (*Ames et al., 1999*; *Liu et al., 1999*; *Mori et al., 1999*; *Nothacker et al., 1999*). UIIs, URPs, and UTs have been linked to cardiovascular function and inflammation, but their roles in the development of morphology are little understood.

In this study, we discovered a role for two of the URPs, Urp1 and Urp2, in zebrafish spine morphology. To do so, we generated $urp1^{\Delta P}$ and $urp2^{\Delta P}$ mutants that lacked the genetic region coding for the Urp1 and Urp2 peptides, respectively, then phenotyped skeletal morphology by μCT. This revealed that Urp1 and Urp2 function semi-redundantly to control spine morphology, with double mutants, as well as Uts2r3 receptor mutants, developing spinal curves in the caudal region during adolescent growth. By contrast, single $urp1^{\Delta P}$ and $urp2^{\Delta P}$ mutants developed more subtle curves that nevertheless worsened with age. The lack of major vertebral defects, the location and direction of the curves coupled with phenotypic differences compared with mutants that model AIS and kyphosis, suggested that urotensin-deficient mutants model a lordosis-like condition.

The RF, a long proteinaceous thread-like structure which sits in the central canal and is mostly made from SCOspondin (encoded by *sspo*), has been implicated in controlling body axis and spine morphology (*Cantaut-Belarif et al., 2018*; *Lu et al., 2020*; *Rose et al., 2020*; *Troutwine et al., 2020*). We find that RF breaks down prior to curve formation in the cilia motility $cfap298^{tm304}$ mutant. Given other studies linking presence of the RF with a linear body axis, this strongly suggests that RF breakdown is a major factor driving spinal curves in $cfap298^{tm304}$ mutants. By contrast, Uts2r3-deficient mutants exhibited an intact RF, demonstrating that curves are not formed by RF breakdown in urotensin pathway mutants and nor do the presence of curves significantly disrupt RF structure. This coheres with a model in which urotensin signals act downstream of RF function in controlling spine morphology. Similarly, *urp1* and *urp2* expression are known to be controlled by RF function during embryonic phases (*Figure 1F*; *Cantaut-Belarif et al., 2020*; *Lu et al., 2020*; *Rose et al., 2020*; *Zhang et al., 2018*).

Intriguingly, motile cilia mutants and *sspo* mutants exhibit three-dimensional spinal curves, with dorso-ventral and medio-lateral curvature (*Grimes et al., 2016*; *Lu et al., 2020*; *Rose et al., 2020*; *Troutwine et al., 2020*). By contrast, we find that urotensin-deficient mutants exhibit largely planar curves, only in the dorso-ventral direction, thereby potentially uncoupling two systems controlling posture (*Picton et al., 2021*). Urp1 and Urp2 are expressed in CSF-cNs (*Figure 1D*; *Quan et al., 2015*), a cell type which consists of both dorsal and ventral subpopulations. While Urp1 and Urp2 are co-expressed in ventral CSF-cNs, only Urp2 is expressed in dorsal CSF-cNs (*Figure 1D*; *Quan et al., 2015*). It is therefore tempting to speculate that dorso-ventral spine shape is mediated specifically by ventral CSF-cNs which express higher amounts of Urp1/Urp2 peptides, resulting in dorso-ventral curves in urotensin-deficient mutants. This also suggests, as above, that while Urp1/Urp2 expression is in part controlled by upstream cilia motility and RF function, decreased urotensin signaling cannot account fully for the spinal curve phenotypes that occur upon loss of cilia motility or the RF. Indeed, the RF is required for both dorsal *and* ventral CSF-cN function (*Orts-Del'Immagine et al., 2020*), which may explain why dorso-ventral and medio-lateral curves occur when RF is disrupted either by cilia motility mutations or mutations to SCOspondin. This scenario is further complicated by the finding that increased *urp1* and *urp2* expression occurs upon mutation of *rpgrip1l*, a gene encoding a component of the ciliary transition zone (*Vesque et al., 2019* — preprint). As such, various cilia-dependent signals likely control the precise levels of Urp1/Urp2 peptides, and controlling those levels appears critical for maintaining the shape of the spine.

In addition to exhibiting phenotypic differences in terms of curve direction compared with cilia motility and RF mutants, urotensin-deficient mutants also showed gradual worsening of spinal phenotypes upon aging. $urp1^{\Delta P}$ mutants exhibited no obvious phenotypes at 3 mpf but by 12 mpf showed abnormal curves, while $urp2^{\Delta P}$ showed mild curves at 3 mpf and more severe deformity at 12 mpf. This implies that Urp1/Urp2 function throughout adulthood and aging to maintain spine morphology. Since double mutants were more severe at 3 mpf than either single mutant, partial redundancy between Urp1 and Urp2 appears to occur while overall dose likely sets how early phenotypes manifest. In contrast to this long-term role for urotensin peptides, temperature-shift experiments in which motile cilia were inactivated after 34 dpf showed no role for motile cilia beyond this stage in maintaining the spine (*Grimes et al., 2016*). It is worth noting, however, that this result does not preclude a role for CSF flow or the RF during long-term spine morphostasis

because it has not been determined if motile cilia are required for CSF flow or RF formation in adult fish.

While the study of zebrafish spinal curve mutants holds great promise for understanding the basic science of spine morphology, it will also be important for the field to grapple with the question of how closely zebrafish spinal deformity mutants truly recapitulate human spinal curve diseases. The spines of humans and zebrafish are broadly similar, and it has been suggested that spinal loads are comparable (*Gorman and Breden, 2009*). Moreover, zebrafish spines, like human spines, seem predisposed to curvature, with high levels of scoliosis-like curves naturally developing with age (*Bearce and Grimes, 2021*; *Gorman and Breden, 2009*). The overall shape of the zebrafish spine is also similar to human, with a natural kyphotic curve in the pre-caudal (rib-bearing) vertebrae and a compensatory, albeit very minor, lordotic curve in the most anterior caudal vertebrae. However, these curves are not as pronounced as in humans. Moreover, zebrafish also exhibit some fish-specific structures such as the Weberian apparatus (*Dietrich et al., 2021*). Nevertheless, zebrafish cilia motility mutants appear to model several features of AIS including the three-dimensional nature of curves, lack of vertebral patterning defects or significant vertebral structural malformations, adolescent-onset and, in some cases, sex bias (*Grimes et al., 2016*; *Marie-Hardy et al., 2021*; *Wang et al., 2022*). The curves of $urp1^{\Delta P};urp2^{\Delta P}$ mutants display some of these features as well but, importantly, are not three-dimensional. Instead, urotensin pathway-deficient mutants display primarily planar curves, with little or no lateral deviation. This is more similar to what occurs in hyper-kyphosis and hyper-lordosis in humans, when natural curves are accentuated. Given this planarity, and since curves are mostly present in the caudal vertebrae, we suggest that $urp1^{\Delta P};urp2^{\Delta P}$ mutants model some aspects of lordosis and so refer to this phenotype as lordosis-like. However, we note that human lumbar vertebrae and zebrafish caudal vertebrae are structurally distinct (*Boswell and Ciruna, 2017*), and humans have a significant natural lordotic curve that allows for an efficient upright walking gait, whereas zebrafish do not. Thus, urotensin-deficient mutants recapitulate some aspects of lordosis but clearly cannot mimic human-specific aspects of hyper-lordotic curves.

A surprising finding from our work was that Urp1 and Urp2 peptides are genetically dispensable for embryonic axial straightening. This interpretation is challenged by morpholino knockdown of Urp1/Urp2, which does result in failure of axial straightening in some individuals, resulting in a CTD phenotype (*Zhang et al., 2018*). One possibility is that the CTD of morphants results from morpholino off-target effects. However, this seems unlikely for three reasons: (1) adding exogenous Urp1/Urp2 peptides to the central canal can rescue the CTD phenotype of cilia motility- and RF-deficient mutants (*Lu et al., 2020*; *Zhang et al., 2018*), suggesting the involvement of Urp1/Urp2 in axial straightening, at least in gain-of-function experiments; (2) since motile cilia and the RF are required for both axial straightening during embryogenesis and for the maintenance of spine morphology during adolescence, it seems parsimonious that Urp1/Urp2 peptides would also function across these two life stages; and (3) *urp1* and *urp2* transcript levels are reduced in motile cilia and RF mutants (*Figure 1F*; *Cantaut-Belarif et al., 2020*; *Lu et al., 2020*; *Zhang et al., 2018*), suggestive of a link between Urp1 and Urp2 upregulation and axial straightening.

Nevertheless, the lack of CTD in our mutants, in which the Urp1 and Urp2 peptide coding sequences were entirely removed, is clear: single, double, and maternal-zygotic $urp1^{\Delta P}$ and $urp2^{\Delta P}$ mutants all underwent normal axial straightening. This strongly argues that Urp1 and Urp2 are dispensable for straightening. Since the deletions were induced toward the end of the protein, it does leave open the possibility that the pro-domain sequences are required for straightening. However, this seems unlikely for three reasons: (1) $urp1^{\Delta P}$ and $urp2^{\Delta P}$ mutants showed *urp1* and *urp2* transcript downregulation, respectively, in addition to deletion of the peptide coding regions; (2) *urp1* and *urp2* single and double crispants, in which gRNAs targeted several regions of the gene, also showed normal straightening; and (3) exogenous addition of Urp1 and Urp2 peptides, without pro-domains, rescued CTD of a cilia motility and RF mutant (*Lu et al., 2020*; *Zhang et al., 2018*), suggesting that it is the peptide itself and not some other region which is functional.

One possibility is that genetic compensation explains the mutant/morphant phenotypic differences (*Rossi et al., 2015*; *Sztal and Stainier, 2020*). In this putative scenario, a feedback response in the cell buffers otherwise harmful mutations, preventing their effects from manifesting phenotypically. A recently discovered compensation mechanism is transcriptional adaptation in which mutant mRNA is decayed and the products of that decay are, via a sequence-dependent mechanism, recruited to

genes with similar sequences where they promote transcriptional upregulation (*El-Brolosy et al., 2019*; *Ma et al., 2019*). The upregulation of adapting genes then masks phenotypes in mutants but not morphants. We did find slight upregulation of *urp2* in *urp1^ΔP* mutants (*Figure 1—figure supplement 2E*), which may indicate transcriptional adaptation, although this alone cannot explain the lack of CTD phenotypes in *urp1^ΔP* mutants because *urp1^ΔP;urp2^ΔP* double mutants also lacked CTD. It will be informative in the future to determine whether other urotensin II-encoding peptides (*Figure 1— figure supplement 1A–B*) are able to compensate, during embryonic phases, for loss of *urp1* and *urp2* or if other factors explain the mutant/morphant discrepancies.

While we were in the final stages of preparing this manuscript, a study was released which made several complementary findings, also concluding that Urp1 and Urp2 function redundantly to maintain spine shape (*Gaillard et al., 2022* — preprint). Importantly, and in agreement with our work, Gaillard and colleagues observed no embryonic axial defects upon genetic loss of Urp1 and Urp2. Moreover, they found that other urotensin II-encoding genes were not upregulated in *urp1* and *urp2* mutants, arguing against phenotypic masking by genetic compensation. Our findings and those of Gaillard and colleagues together therefore strongly argue that Urp1 and Urp2 are not essential for axial straightening during embryogenesis but are instead required for the maintenance of the body axis during growth and adulthood. As such, other mechanisms, currently unknown, likely operate downstream of cilia motility and RF function to mediate embryonic axial straightening.

Future efforts will be required to discern which tissues respond to Urp1/Urp2 signals during the control of spine morphology. During embryonic stages, Uts2r3 is expressed in dorsal muscle (*Zhang et al., 2018*), but it remains unclear how Urp1/Urp2 peptides released by CSF-cNs could signal to effect muscle during adolescent stages. Moreover, based on single-cell RNA-sequencing gene expression atlases (*Farnsworth et al., 2020*), *uts2r3* is expressed in several other cell types in addition to muscle. Tissue-specific ablations and rescue experiments should be used to untangle precisely where Uts2r3-dependent Urp1/Urp2 signaling occurs to control spine morphology. It will also be critical to determine the timing of action of urotensin signaling. While we observe phenotypes first appearing between 9 and 11 dpf in *urp1^ΔP;urp2^ΔP* mutants, it is possible that the underlying defect in mutants is caused by an earlier event which only phenotypically manifests later. One candidate is subtle disruptions to the notochord, which may later result in spinal curves (*Bagwell et al., 2020*). On the other hand, our experiments in which we aged *urp1^ΔP* and *urp2^ΔP* single mutants argue that Urp1 and Urp2 peptides function throughout life, and not only during early growth, to maintain spine morphology.

Another major question is whether the function of urotensin signaling in spine morphology is conserved in other species, and whether our findings are directly relevant to humans. These questions will require future work to answer, but two recent studies shed some light on these matters. First, in the frog *Xenopus laevis*, disruption of Utr4, a counterpart of Uts2r3, causes abnormal curvature of the body axis (*Alejevski et al., 2021*). Second, a human genetics study reported that rare mutations in UTS2R are significantly associated with spinal curvature, being discovered within AIS patient cohorts (*Dai et al., 2021*). Thus, a deeper understanding of the role of urotensin signaling in maintaining spine shape will not only provide insight into principles of morphogenesis but potentially also human disease.

# Materials and methods
## Zebrafish

AB, TU, and WIK strains of *D. rerio* were used. Zebrafish lines generated were *sspo^b1446*, *urp1^b1420* (called *urp1^ΔP*), *urp2^b1421* (called *urp2^ΔP*), *uts2r3^b1436* as well as previously published lines: *cfap298^tm304* (*Jaffe et al., 2016*), *pkd2l1^icm02* (*Sternberg et al., 2018*), and *sspo-gfp^ut24* (*Troutwine et al., 2020*). Experiments were undertaken in accordance with research guidelines of the International Association for Assessment and Accreditation of Laboratory Animal Care and approved by the University of Oregon Institutional Animal Care and Use Committee (Protocol number 21–45). All zebrafish strains and other materials are available upon request.

Generation and genotyping of mutant lines *sspo^b1446*, *urp1^b1420*, *urp2^b1421*, and *uts2r3^b1436* were generated using CRISPR/Cas9. gRNA oligos were designed using CRISPRscan (*Moreno-Mateos et al., 2015*). gRNA templates (IDT) were assembled by annealing and extension with Bottom strand ultramer_1 (Key resource table) using Taq Polymerase (NEB, M0273) with cycling parameters of 95°C

(3 min), 95°C (30 s), 45°C (30 s), 72°C (30 s), and 72°C (10 min) with 30 cycles of the middle three steps. PCR product was purified (Zymo DNA Clean and Concentrator Kit, D4013) and then used for in vitro RNA synthesis using a MEGAshortscript T7 Transcription Kit (ThermoFisher, AM1354). Synthesized gRNAs were purified (Zymo RNA Clean and Concentrator Kit, R1013), then 150 pg along with 320 pg/nl Cas9 (IDT, 1081058) were injected into one-cell stage fertilized eggs. The mosaic mutant fish resulting from these injections ($F_0$ fish) were raised and outcrossed to AB wild-types, and DNA was extracted from the resulting $F_1$ embryos. Mutant alleles were screened by PCR coupled with restriction enzyme digestion and/or Sanger sequencing (GeneWiz). Embryos from $F_1$ clutches harboring mutations were raised to adulthood and outcrossed to AB wild-types to generate $F_2$ families which were screened for mutations and raised. The nature of mutations was identified by sequencing DNA extracted from adult fin clips of $F_2$ heterozygous fish using CRISP-ID to deconvolute (*Dehairs et al., 2016*) and confirmed by sequencing DNA of $F_3$ homozygous embryos.

$urp1^{b1420}$ mutants contain a 279 bp deletion and 1 bp insertion that were genotyped by PCR amplification with *urp1_geno_1* and *urp1_geno_2* primers which generate a 460 bp band from wild-type DNA and a 184 bp band from mutant DNA. $urp2^{b1421}$ mutants harbor a 61 bp deletion and were genotyped by PCR amplification with *urp2_geno_1* and *urp2_geno_2* primers followed by gel electrophoresis to distinguish the 283 bp wild-type band and the 226 bp mutant band. $uts2r3^{b1436}$ mutants contain a 534 bp deletion and were also genotyped by PCR, using *uts2r3_geno_1* and *uts2r3_geno_2*, in which wild-type sequence led to an 832 bp band and mutant sequence a 298 bp band.

The nature of the $sspo^{b1446}$ mutation was determined by whole genome sequencing. DNA was extracted from mutant embryos using a phenol/chloroform procedure. Libraries were prepared using the FS DNA Library Prep Kit for Illumina sequencing (NEB, E7805). DNA was digested into 150 bp fragments, and paired-end sequencing was performed using a NovaSeq 6000 Sequencing System. Trimmomatic (version 0.36; ILLUMNIACLIP: TruSeq3-PE-2.fa:2:30:10:1:true LEADING:3 TRAILING:3 SLIDINGWINDOW:5:20 MINLEN:42 AVGQUAL:30) was used to remove Illumina adaptor sequences from paired-end reads. Illumina short-read sequences were then aligned to the GRCz11 reference sequence of chromosome 24 using BWA-MEM (version 0.7.01). SAMtools (version 1.8) was used to sort and index reads. Aligned reads in BAM format were analyzed in IGV (version 2.13.1). Mutants were routinely genotyped by PCR amplification with oligos *sspo_geno_1* and *sspo_geno_2*, followed by BsaI-HFv2 (NEB, R2733) restriction digestion to produce 300 bp and 99 bp bands from wild-type DNA and a protected 399 bp band from mutant DNA.

## Generation of somatic mosaic $F_0$ mutants (crispants)

Four gRNA oligos per gene for *cfap298*, *sspo*, *urp1*, and *urp2* were chosen from a look-up table (Key resource appendix; *Wu et al., 2018*). gRNAs were synthesized from oligos in multiplex. After being pooled at 10 µM, oligos were annealed and extended with Bottom strand ultramer_2 using Phusion High-Fidelity PCR Mastermix (NEB, M0531) with Phusion High-Fidelity DNA Polymerase (NEB, M05030) using incubations: 98°C (2 min), 50°C (10 min), and 72°C (10 min). Assembled oligos were purified and used as templates for in vitro RNA synthesis, as described in 'Generation and genotyping of mutant lines' section. For mutagenesis, 1000 pg of gRNAs along with 1600 pg/nl Cas9 (IDT, 1081058) were injected into one-cell-stage embryos. To assess rates of mutagenesis, DNA was extracted from 1 dpf crispants and subjected to T7 endonuclease I assays (NEB, E3321).

## Quantitation of body curvature at 1–2 dpf

Zebrafish larvae at 28–30 hpf were imaged using a Leica S9i stereomicroscope with integrated 10-megapixel camera. Body angles were calculated using ImageJ (*Schindelin et al., 2012*) as described in *Bearce et al., 2022*.

## Quantitative reverse transcriptase PCR (qRT-PCR)

RNA was extracted using a Zymo Direct-Zol RNA Miniprep kit (Zymo Research, R2051). cDNA was prepared from 25 ng of RNA using oligoDT primers in a 20 µl reaction using a High Capacity cDNA Reverse Transcription Kit (ThermoFisher, 4368814). qRT-PCR reactions were performed in real time using 5 µl PowerUp SYBR Green Master Mix (ThermoFisher, A25741), 0.8 µl of 10 µM forward and reverse primers, 1.4 µl of nuclease-free water, and 2 µl of diluted cDNA. PCR was performed using a QuantStudio Real Time PCR System (Applied Biosystems) with cycling parameters: 50°C (2 min), 95°C

(10 min) then 40 cycles of 95°C (15 s), and 60°C (1 min). Each reaction was performed in quadruplicate. Quantitation was relative to *rpl13* and used the $\Delta\Delta C_T$ relative quantitation method in which fold changes are calculated as $2^{-\Delta\Delta CT}$. The efficiency of amplification was verified to be close to 100% with a standard curve of RNA dilutions.

### Calcein staining
Larvae were incubated in water containing 0.2% calcein (Sigma-Aldrich, C0875) for 10 min then rinsed two to three times in water (5 min per rinse). Larvae were immobilized with 0.005% tricaine, mounted in 0.8% low melt agarose, and imaged with a Leica THUNDER stereoscope.

### Multiplex fluorescent in situ hybridization chain reaction (in situ HCR)
Embryos were fixed in 4% paraformaldehyde at 4°C overnight, washed with phosphate buffered saline (PBS) then serially dehydrated to 100% methanol, and stored at –20°C. Embryos were rehydrated, washed with PBS containing 0.1% Tween-20, incubated in hybridization buffer (Molecular Instruments), then incubated in 2 pmol of probes at 37°C overnight in a total volume of 500 µl of hybridization buffer. Embryos were washed in wash buffer (Molecular Instruments), washed twice in 5× SSCT (sodium chloride sodium citrate with 0.1% tween-20), and then incubated in amplification buffer (Molecular Instruments) for 1 hr. RNA hairpins designed to bind either *pkd2l1*, *urp1*, or *urp2* were prepared by heating 10 pmol of each to 95°C for 90 s then snap-cooled in the dark for 30 min. Embryos were then incubated overnight in 500 µl of amplification buffer containing 30 pmol hairpins at room temperature in the dark. Embryos were washed five times in 5× SSCT, stored at 4°C, and then mounted for confocal microscopy. Images were acquired using a Zeiss LSM880 using either a ×20 air or ×40 water objective. Acquisition settings were derived using wild type embryos and then applied to all embryos. Images were exported to IMARIS 9.5.0 (Oxford Instruments). A Gaussian filter of width 0.42 µm (×20) or 0.21 µm (×40), and a rolling ball background subtraction of 10 µm was applied.

### X-ray microcomputed tomography
Scans were performed using a vivaCT 80 (Scanco Medical) at 18.5-µm voxel resolution (for 3 mpf and 12 mpf fish) or 10-µm voxel resolution (1 mpf fish) as previously described (*Bearce et al., 2022*). Digital dissections of the spinal column were performed in 3D Slicer (*Kikinis et al., 2013*) using the Segmentation Editor. A threshold of 3200 was used to mask 1 mm tube (Draw Tube function) around the spine in the axial slice view, beginning between the otic vesicles rostral to the first vertebrae and ending at the split of the tail. The center of the tube was set at the narrowest 'hollow'' within the lumen of centra.

### Quantitation of lateral spine curvature
Quantitation of lateral spine curvature was performed in ImageJ by orienting isolated spine images in a dorsal view with heads to the left and with the otic vesicles and first Weberian vertebrae parallel to the x-axis. Landmarks were assigned to the narrowest point of each centrum rostral to caudal; where maximum projection resulted in hidden or overlapping vertebrae, the appropriate number of points was added in closest approximation. The y-value from each landmark was subtracted from the point rostral to it, resulting in a map of local deflections where positive values indicate rightward displacement, and negative values indicate leftward displacement.

### Live imaging of SCOspondin-GFP
Embryos (28 hpf) and larvae (12 dpf) were anesthetized in tricaine until touch response was abolished and then embedded in 0.8% low-melt agarose laced with tricaine in inverted imaging chambers (14 mm #1.5 coverslips, VWR cat no. 10810–054). In larvae, care was taken to align the posterior body close to the coverslip to the RF within the working distance of the objective. A Nikon Ti2 inverted microscope equipped with Plan Apo ×40 and ×60 WI DIC (1.2 NA) objectives, a Yokogawa Spinning Disk and pco.edge sCMOS camera were used to capture 512×256 images in time series. Exposure time varied with age (100 ms–300 ms) as Sspo-GFP brightened in intensity over time; exposure, camera settings, and laser power were kept constant between age-matched individuals. Images were cropped, rotated, and intensity-adjusted in ImageJ (*Schindelin et al., 2012*).

## Acknowledgements

We thank Judy Peirce, Tim Mason, and the Aquatics Facility for zebrafish husbandry, the GC3F Biological Imaging Facility, and the X-Ray Imaging Core, all at the University of Oregon. We thank Claire Wyart, John Postlethwait, and Ryan Gray for sharing zebrafish lines, Zac Bush for help with genome sequencing, Mike Harms and Ron Kwon for discussions, and Katie Fisher for proof reading. Funding: This study was supported by a National Institutes of Health grants R00AR70905 and R35GM142949 (to DTG), F32AR078002 (to EAB), F31HD105435 and T32HD007348 (to ZHI).

## Additional information

### Funding

| Funder | Grant reference number | Author |
| --- | --- | --- |
| National Institutes of Health | R00AR70905 | Daniel T Grimes |
| National Institutes of Health | F32AR078002 | Elizabeth A Bearce |
| National Institutes of Health | F31HD105435 | Zoe H Irons |
| National Institutes of Health | R35GM142949 | Daniel T Grimes |

The funders had no role in study design, data collection and interpretation, or the decision to submit the work for publication.

### Author contributions

Elizabeth A Bearce, Formal analysis, Funding acquisition, Investigation, Visualization, Methodology, Writing - review and editing; Zoe H Irons, Funding acquisition, Validation, Methodology; Johnathan R O'Hara-Smith, Investigation, Visualization, Methodology, Writing - review and editing; Colin J Kuhns, Validation, Investigation; Sophie I Fisher, William E Crow, Investigation; Daniel T Grimes, Conceptualization, Supervision, Funding acquisition, Investigation, Writing - original draft, Project administration, Writing - review and editing

### Author ORCIDs

William E Crow (i) http://orcid.org/0000-0003-2991-3076
Daniel T Grimes (i) http://orcid.org/0000-0003-0173-1887

### Ethics

Experiments were undertaken in accordance with research guidelines of the International Association for Assessment and Accreditation of Laboratory Animal Care and approved by the University of Oregon Institutional Animal Care and Use Committee (# AUP-21-45).

### Decision letter and Author response

Decision letter https://doi.org/10.7554/eLife.83883.sa1
Author response https://doi.org/10.7554/eLife.83883.sa2

## Additional files

### Supplementary files

- MDAR checklist

### Data availability

All data generated or analysed during this study are included in the manuscript and supporting file.

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

# Appendix 1

## Appendix 1—key resources table

| Reagent type (species) or resource | Designation | Source or reference | Identifiers | Additional information |
|---|---|---|---|---|
| Commercial assay, kit | DNA Clean and Concentrator Kit | Zymo Research | Cat no: D4013 | |
| Commercial assay, kit | RNA Clean and Concentrator Kit | Zymo Research | Cat no: R1016 | |
| Commercial assay, kit | Direct-zol RNA MiniPrep Kit | Zymo Research | Cat no: R2050 | |
| Commercial assay, kit | GeneJET Gel Extraction Kit | Thermo Fisher Scientific | Cat no: K0691 | |
| Commercial assay, kit | High Capacity RNA-to-cDNA Kit | Thermo Fisher Scientific | Cat no: 4387406 | |
| Commercial assay, kit | MEGAshortscript T7 Transcription Kit | Thermo Fisher Scientific | Cat no: AM1354 | |
| Commercial assay, kit | HiScribe T7 High Yield RNA Synthesis Kit | New England Biolabs | Cat no: E2040 | |
| Commercial assay, kit | FS DNA Library Prep Kit for Illumina | New England Biolabs | Cat no: E7805 | |
| Commercial assay, kit | HCR-RNA FISH Hybridization Buffer | Molecular Instruments | | |
| Commercial assay, kit | T7 Endonuclease I | New England Biolabs | Cat no: E3321 | |
| Commercial assay, kit | HCR-RNA FISH Amplification Buffer | Molecular Instruments | | |
| Commercial assay, kit | AlexaFluor-647 Hairpins | Molecular Instruments | | |
| Commercial assay, kit | AlexaFluor-546 Hairpins | Molecular Instruments | | |
| Commercial assay, kit | AlexaFluor-488 Hairpins | Molecular Instruments | | |
| Commercial assay, kit | *urp1* HCR-RNA FISH probe | Molecular Instruments | | Project-specific design |
| Commercial assay, kit | *urp2* HCR-RNA FISH probe | Molecular Instruments | | Project-specific design |
| Commercial assay, kit | *pkd2l1* HCR-RNA FISH probe | Molecular Instruments | | Project-specific design |
| Commercial assay, kit | Taq Polymerase | New England Biolabs | Cat no: M0273 | |
| Commercial assay, kit | TURBO DNase | Thermo Fisher Scientific | Cat no: AM2238 | |
| Commercial assay, kit | Phusion High-Fidelity DNA Polymerase | New England Biolabs | Cat no: M0530 | |
| Commercial assay, kit | Phusion High-Fidelity PCR Master Mix | New England Biolabs | Cat no: M0531 | |
| Commercial assay, kit | SYBR Green PCR Master Mix | Thermo Fisher Scientific | Cat no: 4309155 | |
| Strain, strain background (*Danio rerio*) | AB strain | University of Oregon | | |

*Appendix 1 Continued on next page*

*Appendix 1 Continued*

| Reagent type (species) or resource | Designation | Source or reference | Identifiers | Additional information |
|---|---|---|---|---|
| Strain, strain background (*D. rerio*) | WIK strain | University of Oregon | | |
| Strain, strain background (*D. rerio*) | TU strain | University of Oregon | | |
| Strain, strain background (*D. rerio*) | cfap298<sup>tm304</sup> line | *Jaffe et al., 2016* | ZDB-FISH-150901–23024 | |
| Strain, strain background (*D. rerio*) | pkd2l1<sup>icm02</sup> line | *Sternberg et al., 2018* | ZDB-FISH-160811–9 | |
| Strain, strain background (*D. rerio*) | sspo<sup>b1446</sup> line | This study | | *Figure 1—figure supplement 3* |
| Strain, strain background (*D. rerio*) | sspo-GFP<sup>ut24</sup> line | *Troutwine et al., 2020* | ZDB-FISH-190313–20 | |
| Strain, strain background (*D. rerio*) | urp1<sup>b1420</sup> line | This study | | *Figure 1—figure supplement 2* |
| Strain, strain background (*D. rerio*) | urp2<sup>b1421</sup> line | This study | | *Figure 1—figure supplement 2* |
| Strain, strain background (*D. rerio*) | uts2r3<sup>b1436</sup> line | This study | | *Figure 2—figure supplement 3* |
| Software, algorithm | 3D Slicer | *Fedorov et al., 2012* | | |
| Software, algorithm | IMARIS 9.9 | Oxford Instruments | | |
| Software, algorithm | NIS-Elements | Nikon Instruments Inc | | |
| Software, algorithm | ZEN Software | Carl Zeiss AG | | |
| Software, algorithm | QuantStudio Design and Analysis Software | Applied Biosystems | | |
| Software, algorithm | Integrated Genome Viewer (version 2.13.1) | *Robinson et al., 2011* | | |
| Sequence-based reagent | Bottom strand ultramer_1 | This study | Tail ultramer for generating singe gRNA oligos | AAAAGCACCGACTCG GTGCCACTTTTTC AAGTTGATAACGGACT AGCCTTATTTT AACTTGCTAT |
| Sequence-based reagent | *urp1_gRNA_1* | This study | gRNA_1 oligo for generating urp1<sup>b1420</sup> line | taatacgactcactataGGCGT TGGTCAGCCTGACAT gttttagagctagaa |
| Sequence-based reagent | *urp1_gRNA_2* | This study | gRNA_2 oligo for generating urp1<sup>b1420</sup> line | taatacgactcactataGGG TCCTCTGTCCATCTCCG gttttagagctagaa |
| Sequence-based reagent | *urp2_gRNA_1* | This study | gRNA_1 oligo for generating urp2<sup>b1421</sup> line | taatacgactcactataGGCA GATGGAGAAAGATTGA gttttagagctagaa |

*Appendix 1 Continued on next page*

*Appendix 1 Continued*

| Reagent type (species) or resource | Designation | Source or reference | Identifiers | Additional information |
|---|---|---|---|---|
| Sequence-based reagent | *urp2_gRNA_2* | This study | gRNA_2 oligo for generating *urp2*[b1421] line | taatacgactcactataGGC GTTTGCAGAAATCAGCG gttttagagctagaa |
| Sequence-based reagent | *uts2r3_gRNA* | This study | gRNA oligo for generating *uts2r3*[b1436] line | taatacgactcactataGGG TGAAGGGGAAGAGAAGA gttttagagctagaa |
| Sequence-based reagent | *sspo_gRNA_1* | This study | gRNA_1 oligo for generating *sspo*[b1446] line | taatacgactcactataGGT CCCCAGTGGTCCGCG GTgttttagagctagaa |
| Sequence-based reagent | *sspo_gRNA_2* | This study | gRNA_2 oligo for generating *sspo*[b1446] line | taatacgactcactataGGCAC AGTGTGTGAGACCAG gttttagagctagaa |
| Sequence-based reagent | Bottom strand ultramer_2 | This study | Tail ultramer for generating multiplexed gRNA oligos | AAAAGCACCGACTCGG TGCCACTTTTTC AAGTTGATAA CGGACTAGCCTTATT TTAACTTGCTATTTC TAGCTCTAAAAC |
| Sequence-based reagent | urp1_F0_gRNA_1 | *Wu et al., 2018* | gRNA_1 oligo for generating *urp1* F0 embryos | TAATACGACTCACTAT AGGAAAGTGAAGAT CGCGGCCGTTTTAG AGCTAGAAATAGC |
| Sequence-based reagent | urp1_F0_gRNA_2 | *Wu et al., 2018* | gRNA_2 oligo for generating *urp1* F0 embryos | TAATACGACTCACT ATAGGACACGG CTCTGCC ACAACGTTTTAGA GCTAGAAATAGC |
| Sequence-based reagent | urp1_F0_gRNA_3 | *Wu et al., 2018* | gRNA_3 oligo for generating *urp1* F0 embryos | TAATACGACTCACTA TAGGTTCAGAAGC TGGTAGCAGGTTTTA GAGCTAGAAATAGC |
| Sequence-based reagent | urp1_F0_gRNA_4 | *Wu et al., 2018* | gRNA_4 oligo for generating *urp1* F0 embryos | TAATACGACTCACT ATAGGGAAAATAAAT AACATGGTGTTTTA GAGCTAGAAATAGC |
| Sequence-based reagent | urp2_F0_gRNA_1 | *Wu et al., 2018* | gRNA_1 oligo for generating *urp2* F0 embryos | TAATACGACTCACTA TAGGTGACTGTCGC TTCAATCGGTTTTAG AGCTAGAAATAGC |
| Sequence-based reagent | urp2_F0_gRNA_2 | *Wu et al., 2018* | gRNA_2 oligo for generating *urp2* F0 embryos | TAATACGACTCACT ATAGGGACATTTCCT GACGGAGAGTTTTA GAGCTAGAAATAGC |
| Sequence-based reagent | urp2_F0_gRNA_3 | *Wu et al., 2018* | gRNA_3 oligo for generating *urp2* F0 embryos | TAATACGACTCACTA TAGGTGGACACGA GGAGACCGAGTTTT AGAGCTAGAAATAGC |
| Sequence-based reagent | urp2_F0_gRNA_4 | *Wu et al., 2018* | gRNA_4 oligo for generating *urp2* F0 embryos | TAATACGACTCACT ATAGGTCACCAGGTAG TGACGGAGTTTTAG AGCTAGAAATAGC |
| Sequence-based reagent | sspo_F0_gRNA_1 | *Wu et al., 2018* | gRNA_1 oligo for generating *sspo* F0 embryos | TAATACGACTCACT ATAGGTTCGTCCCC AGTGGTCCGGTTTT AGAGCTAGAAATAGC |
| Sequence-based reagent | sspo_F0_gRNA_2 | *Wu et al., 2018* | gRNA_2 oligo for generating *sspo* F0 embryos | TAATACGACTCACT ATAGGAAACGG CCGTCAGTGTCGGT TTTAGAGCTAGAAATAGC |

*Appendix 1 Continued on next page*

*Appendix 1 Continued*

| Reagent type (species) or resource | Designation | Source or reference | Identifiers | Additional information |
|---|---|---|---|---|
| Sequence-based reagent | sspo_F0_gRNA_3 | *Wu et al., 2018* | gRNA_3 oligo for generating *sspo* F0 embryos | TAATACGACTCAC TATAGGTGTTGC AACACCAACCGGGT TTTAGAGCTAGAAATAGC |
| Sequence-based reagent | sspo_F0_gRNA_4 | *Wu et al., 2018* | gRNA_4 oligo for generating *sspo* F0 embryos | TAATACGACTCACT ATAGGAGCCTAGACC TGCTCACGGTTTTA GAGCTAGAAATAGC |
| Sequence-based reagent | cfap298_F0_gRNA_1 | *Wu et al., 2018* | gRNA_1 oligo for generating *cfap298* F0 embryos | TAATACGACTCAC TATAGGTTCTCTT CAACACTACGGGT TTTAGAGCTAGAAATAGC |
| Sequence-based reagent | cfap298_F0_gRNA_2 | *Wu et al., 2018* | gRNA_2 oligo for generating *cfap298* F0 embryos | TAATACGACTCAC TATAGGGCTCC ACAATCTGATCATG TTTTAGAGCTA GAAATAGC |
| Sequence-based reagent | cfap298_F0_gRNA_3 | *Wu et al., 2018* | gRNA_3 oligo for generating *cfap298* F0 embryos | TAATACGACTCAC TATAGGCATTC TTATTGGATCATGG TTTTAGAGCTA GAAATAGC |
| Sequence-based reagent | cfap298_F0_gRNA_4 | *Wu et al., 2018* | gRNA_4 oligo for generating *cfap298* F0 embryos | TAATACGACTCACT ATAGGTCTCTGG CAGGTGCGCCCGTT TTAGAGCTAGAAATAGC |
| Sequence-based reagent | pkd2l1_geno_1 | *Sternberg et al., 2018* | *pkd2l1^{icm02}* genotyping oligo 1 | TGTGTGCTAGG ACTGTGGGG |
| Sequence-based reagent | pkd2l1_geno_2 | *Sternberg et al., 2018* | *pkd2l1^{icm02}* genotyping oligo 2 | AGGGCAAGAGAA TGGCAAGACG |
| Sequence-based reagent | urp1_geno_1 | This Study | *urp1^{b1420}* genotyping oligo 1 | GCACCCAAAAT CCAACGACT |
| Sequence-based reagent | urp1_geno_2 | This Study | *urp1^{b1420}* genotyping oligo 2 | TGTATGGGGAA AACAAAGGCA |
| Sequence-based reagent | urp2_geno_1 | This Study | *urp2^{b1421}* genotyping oligo 1 | TTGGGGTTGT AACAGGTAGTG |
| Sequence-based reagent | urp2_geno_2 | This Study | *urp2^{b1421}* genotyping oligo 2 | AACAAGGAAGA CGCTGCAAG |
| Sequence-based reagent | uts2r3_geno_1 | This Study | *uts2r3^{b1436}* genotyping oligo 1 | ATGGATCCCC TGATGTCCTG |
| Sequence-based reagent | uts2r3_geno_2 | This Study | *uts2r3^{b1436}* genotyping oligo 2 | TCGAACTCTGC TCATCCCAG |
| Sequence-based reagent | *sspo_geno_1* | This Study | *sspo^{b1446}* genotyping oligo 1 | CGCAAACACTT CCACTTCCA |
| Sequence-based reagent | *sspo_geno_2* | This Study | *sspo^{b1446}* genotyping oligo 2 | TTGAAGCCAGATGT AAAGGATGAGTGT |
| Sequence-based reagent | *urp1_gRNA1+2_T7E1_F* | This Study | Forward primer to amplify genomic DNA for T7E1 assay in crispants | GACAGCGCAC CCTTAATTGT |
| Sequence-based reagent | *urp1_gRNA1+2_T7E1_R* | This Study | Reverse primer to amplify genomic DNA for T7E1 assay in crispants | ACATTTAGCCTT AACAAGCACAA |

*Appendix 1 Continued on next page*

Appendix 1 Continued

| Reagent type (species) or resource | Designation | Source or reference | Identifiers | Additional information |
|---|---|---|---|---|
| Sequence-based reagent | urp1_gRNA3+4_T7E1_F | This Study | Forward primer to amplify genomic DNA for T7E1 assay in crispants | CAGACAAGGG AACAGAGAGGA |
| Sequence-based reagent | urp1_gRNA3+4_T7E1_R | This Study | Reverse primer to amplify genomic DNA for T7E1 assay in crispants | CCACTGCTTTTA AATCATCCACC |
| Sequence-based reagent | urp2_gRNA1_T7E1_F | This Study | Forward primer to amplify genomic DNA for T7E1 assay in crispants | ATCTTAGAGG CGCATTGGTG |
| Sequence-based reagent | urp2_gRNA1_T7E1_R | This Study | Reverse primer to amplify genomic DNA for T7E1 assay in crispants | GCATGAGGCG GTTTGTTTTG |
| Sequence-based reagent | urp2_gRNA2+3_T7E1_F | This Study | Forward primer to amplify genomic DNA for T7E1 assay in crispants | TGAAGCAACT GAGGAGCAAA |
| Sequence-based reagent | urp2_gRNA2+3_T7E1_R | This Study | Reverse primer to amplify genomic DNA for T7E1 assay in crispants | ACAGTACAGTT CAGCACACCT |
| Sequence-based reagent | urp2_gRNA4_T7E1_F | This Study | Forward primer to amplify genomic DNA for T7E1 assay in crispants | TGACCTATACATC AAAGCCAAGG |
| Sequence-based reagent | urp2_gRNA4_T7E1_R | This Study | Reverse primer to amplify genomic DNA for T7E1 assay in crispants | CCTGGGCTGA TCATACCTCT |
| Sequence-based reagent | rpl13_qPCR_F | This Study | Forward primer for quantitative RT-PCR | TAAGGACGGAG TGAACAACCA |
| Sequence-based reagent | rpl13_qPCR_R | This Study | Reverse primer for quantitative RT-PCR | CTTACGTCTGC GGATCTTTCTG |
| Sequence-based reagent | urp1_qPCR_F | This Study | Forward primer for quantitative RT-PCR | ACATTCTGG CTGTGGTTTG |
| Sequence-based reagent | urp1_qPCR_R | This Study | Reverse primer for quantitative RT-PCR | GTCCGTCTTCA ACCTCTGCTAC |
| Sequence-based reagent | urp2_qPCR_F | This Study | Forward primer for quantitative RT-PCR | AGAGGAAACA GCAATGGACG |
| Sequence-based reagent | urp2_qPCR_R | This Study | Reverse primer for quantitative RT-PCR | TGTTGGTTTTG GTTGACG |

