## [Editor Report]

This is a beautifully executed study on the role of Urp signaling in spine morphogenesis in zebrafish. This work challenges the model that Urp1/ 2 controls the extension and straightening of the body axis of the zebrafish embryos, and thus makes a significant contribution to the literature.

---

## [Decision Letter]

[Editors' note: this paper was reviewed by Review Commons.]

---

## [Author Response]

General Statements

We were immensely pleased that the reviewers considered our conclusions “well supported” and our study “beautifully executed”. Reviewers also recognized the significance of our work. Reviewer 1 stated that “building a model that describes one of these pathways will allow us to begin to test therapies to treat or prevent scoliosis” then noted that we “help to build a larger model of normal spine morphogenesis” and that this is “important”. Reviewer 2 called our work an “exciting advance in our understanding of one of the essential signaling pathways that help regulate body axis straightening and spine morphogenesis in zebrafish” and mentioned that our work “may also help to further our understanding of the etiology and pathophysiology of multiple forms of neuromuscular scoliosis in humans”. Reviewer 3 agreed, stating that our work “adds important information on the role of urotensin signaling in spine formation” and noted that it is timely: “findings are of special significance in the light of recent reports that mutations in UTS2R3 show association with spinal curvature in patients with adolescent idiopathic scoliosis”.

We thank the three reviewers for reading our research and providing feedback. In all cases, we have incorporated their suggestions, and we believe this has made our manuscript much stronger. Indeed, reviewers had only a small number of “major points”, and all have now been addressed. We have also addressed all “minor points” raised by reviewers.

Major pointsReviewer 1The key conclusions are well supported, see below for my two major issues.1. Please don't call this lordosis. Lordosis or hyperlordosis effects lumbar vertebra. The curve in the lumbar region shifts body weight so that human gait is more efficient that in the great apes, or so the story goes. Zebrafish do not have lumbar vertebra equivalents or a natural curve in the caudal region. Similarly, fish do not have the equivalent vertebra to generate kyphosis, which is again a hyper flexion of a normal human spinal curve. Instead zebrafish have Weberian, precaudal and caudal vertebra. It would be so much more useful for the field if the authors used these terms and specified ranges, i.e. numbered vertebrae, that are effected so we can directly and accurately compare regions of defects between zebrafish mutants. It would help to make the point that the uts2r3 mutant has more caudally located curves than urp1/2 double mutants.

We agree with the reviewer and have made changes in line with their suggestions. For example, at the end of the abstract, we have changed “new animal models of lordosis-like curves” to “new animal models of spine deformity”; at the end of the Introduction, we say “mutants model human spinal deformity” rather than “mutants model a lordosis-like condition”.

We have also added a new paragraph to the Discussion about whether zebrafish can truly model scoliosis, lordosis and kyphosis which, among other points, mentions some of what the reviewer brought up about lumbar vs. caudal vertebral differences.

When we do mention lordosis (less often than previously) in the text, we are careful to say “lordosis-like” and we fully explain what we mean by this, including the aspects of the zebrafish phenotypes that are not lordosis-like. Last, when describing our phenotypes, we now use the appropriate names for zebrafish vertebrae, i.e., caudal rather than lumbar.

2. The observation that urp1/2 double mutants have curves only in the D/V plane and almost completely lack side-to-side curves is noteworthy. Does the urp1-/-urp2-/- mutant uncouple two systems for posture? If this separate a DV from side-to-side postural control system, that would be very interesting. It is particularly important to describe how penetrant the phenotype is and how many times it was observed. See 9 minor comments. It would help the reader if the authors explicitly described the features that they see in the cfap298 mutant that constitute lateral curves and that are lacking in urp1/2 (e.g. in figure 4E).

To address this comment, we have supplied more data on the extent of lateral curvature in *cfap298^tm304^* and *urp1^∆P^;urp2^∆P^* mutants and, in the new version, also *uts2r3^b1436^* and *cfap298^tm304^;urp1^∆P^;urp2^∆P^* triple mutants. These new data are:

New data showing the extent of lateral curvature in various mutant conditions (manuscript *Figure 4Fi-Fii*).

Additionally, in the interests of making all of our data available, we provide a supplementary figure showing all scans performed that were used to analyze lateral curvature where we also note the sex of the fish. See new *Figure 4—figure supplement 1*.

In agreement with our previous observations, *cfap298^tm304^* mutants indeed exhibit significantly more lateral curvature than *urp1^∆P^;urp2^∆P^* or *uts2r3^b1436^* mutants. Thus, these mutants may uncouple two systems for posture, as the reviewer suggests. We now note this possibility in the Discussion.

Reviewer 21. Need to show that the CRISPANT targeting was effective for mutagenesis at each loci screened in the work presented in Figure 1E.

We have now performed T7 endonuclease assays (as also suggested by Reviewer 3) to assess mutagenesis in crispants. These assays revealed significant insertion-deletion mutations at all gRNA targeting sites as well as cases of large deletions between sites.

New data demonstrating that *urp1* and *urp2* targeting was effective for mutagenesis (see also *Figure 1—figure supplement 2C-D*):

These data are referenced in the main text in the following way: “Using T7 endonuclease assays, we confirmed that high levels of insertion-deletion mutations were generated at gRNA sites in crispants (*Figure 1—figure supplement 2C-D*).”

Overall, the evidence we have that Urp1 and Urp2 peptides do not contribute to axial straightening is:

*urp1^∆P^* and *urp2^∆P^* single and double mutants (and maternal-zygotic mutants) show normal axial straightening. These mutants completely lack the coding region for the Urp1 and Urp2 peptides.*urp1* and *urp2* crispants with significant indels (as assessed by T7E1 assays) also showed normal straightening. By contrast, all other positive control crispants showed expected axial phenotypes.A preprint produced independently of our work also generated *urp1* and *urp2* mutants and also reported lack of axial defects (Gaillard et al., 2022 – *BioRxiv*).

Thus, we conclude that Urp1 and Urp2 are dispensable for axial straightening in zebrafish. We devote several paragraphs in our Discussion to this somewhat unexpected finding.

Reviewer 31. The addition of the F0 crispant experiment to show that the pro-peptide of urp1/2 does not have a function and is responsible for the difference between the observed morpholino and the crispr phenotype was important. However, since no phenotype was observed in crispants it is important to add evidence of induced cuts for all guide RNAs used in the crispant experiment. These control experiments might have been done already. If not, they can easily be done in a short period of time by performance of T7 assays on injected fish and would not require additional reagents.

Please see response to Reviewer 2 above including Figure1-figure supplement 2. In summary, we have now performed the suggested T7 assays and they indicated that indels were indeed generated in *urp1* and *urp2* crispants (in the manuscript, now described in *Figure 1—figure supplement 2C-D*).

Overall, the evidence we have that Urp1 and Urp2 peptides do not contribute to axial straightening is:

*urp1^∆P^* and *urp2^∆P^* single and double mutants (and maternal-zygotic mutants) show normal axial straightening. These mutants completely lack the coding region for the Urp1 and Urp2 peptides.*urp1* and *urp2* crispants with significant indels (as assessed by T7E1 assays) also showed normal straightening. By contrast, all other positive control crispants showed expected axial phenotypes.A preprint produced independently of our work also generated *urp1* and *urp2* mutants and also reported lack of axial defects (Gaillard et al., 2022 – *BioRxiv*).

2. The authors claim that there were no structural defects observed in urp1/2 double mutants. However, the hemal arch in figure 3 E seems to be deformed. This could be normal variance or a phenotype. This can be addressed by simple reinspection of the scans.

Reinspection of the scans shows no significant hemal arch defects in *urp1^∆P^;urp2^∆P^* mutants. We now explicitly state this in the text. We also note that we do not see significant vertebral defects such as fusions or hemivertebrae and nor do we see any patterning defects at 10 dpf based on calcein staining. Additionally, we have added vertebral body shape quantitation (length and height aspect ratios):

New data showing vertebral body shape quantitation (see also *Figure 3F*)

Since these structural changes are subtle, and we do not observe vertebral patterning changes based on calcein staining at earlier time points, we conclude that the vertebral structural changes are not causative in spinal curvature but rather are a consequence of the presence of curves.

Minor pointsReviewer 13. Supplementary FigS3B How to measure the Cobb Angle is unclear. Why is the first curve not counted? I count 3 curves. First a ventral displacement, then a dorsal to ventral return, then a sharp flex before the tail. How to measure Cobb angle might be easier to explain if the figure is expanded into steps. Identify the apical vertebra, then showing how the lines are drawn parallel to those vertebrae, then where the measured angle forms between the lines perpendicular to the drawn parallel lines.

We have now redrawn our schematic figure more thoroughly explained how Cobb angle is measured with the following figure and figure legend.

5a. I think we (zebrafish biologists) need be explicit about what we mean with "without vertebral defects." What do we count as defects? Vertebrae can be fused, bent, shortened or the growing edges can be slanted. In Figure 3E, and movie7, it is clear that the highlighted mutant vertebrae are shorter than WT. The growing ends of normal vertebra are perpendicular to the long axis of the vertebra. In the mutants the ends are slanted. Please define in the text what you consider a relevant vertebral defect, because these vertebrae have defects. Or are you only considering the calcein stained centra at 10dpf?

We have made textual edits to clearly state that we do not see vertebral fusions or missing appendages. We have also added vertebral body length and height aspect ratios to quantify vertebrae shape and find subtle changes in the variance of vertebral body length aspect ratios in *urp1^∆P^;urp2^∆P^* mutants (see *Figure 3F* in the new manuscript). Since these structural changes are subtle, and we do not observe vertebral patterning changes based on calcein staining at earlier time points, we conclude that the vertebral structural changes are not causative in spinal curvature but rather are a consequence of the presence of curves.

5b. Do you want to base your patterning conclusion on primarily the calcein data as these are closer to the notochord patterning time window. Please anchor this conclusion to a specific time or standard length e.g. 10dpf/5.6mm.

We now note the time window when calcein was performed and provide the standard lengths of fish analyzed. As such, our conclusion about vertebral patterning is now anchored to the standard length, as the reviewer suggested:

“Staining of juveniles with the vital dye calcein revealed no defects in vertebral patterning or spacing in *urp1^∆P^;urp2^∆P^* mutants at 10 dpf (4.5-6 mm standard length [*Figure 3C*]) (*Figure 3F*).”

6. "At 30 dpf… several mutants exhibited a significant curve in the pre-caudal vertebrae, in addition to a caudal curve (Figure 3D and S3C). Since pre-caudal curves were rare in mutants at 3-months, this suggested that curve location is dynamic". The frequency of this observation is important. Does it effect all or a fraction of mutants? Can you provide some numbers to anchor these observations? Maybe fractions e.g.. 3 of 4 fish had precaudal curves at 30pdf, and 0 of 10 fish had precaudal curves by 3 mpf?

We have now analyzed more 1 mpf fish by µCT and we observe that 5 of 7 exhibit pre-caudal curves:

New data showing spinal curves are variable in 1 mpf *urp1^∆P^;urp2^∆P^* mutants (see also *Figure 3—figure supplement 1*).

These data are also mentioned in our revised text. By contrast to the significant pre-caudal curves at 1 mpf, 3 mpf fish showed predominantly caudal curves (*Figure 2F*). Only 2 of 9 fish analyzed by µCT at 3 mpf were scored as exhibiting an “apex of curve” vertebrae in the precaudal vertebrae, and in both of those cases the curve was at the very end of the precaudals. This reinforces our conclusion that curve position is dynamic through growth.

7. The description of the pkd2l1 mutant, instead of terming it kyphosis can you tell the reader the vertebra number at the peak of the curve. The authors say the pkd2l1 mutant is highly distinct from urp1/urp2-/-, but the reader needs to hear exactly what is distinct. For example, does this mutant have both lateral and D/V curves?

We have now scanned several *pkd2l1^icm02^* mutant fish include images of *pkd2l1^icm02^* mutants at two different timepoints (3 mpf and 12 mpf; see new *Figure 2—figure supplement 2*). Our results agreed with those previously published for the *pkd2l1^icm02^* mutant line (Sternberg et al., 2018) but we believe it is important to include our data so readers can see side-by-side images of the various mutant conditions using the same skeleton visualization technique.

At 3 mpf, *pkd2l1^icm02^* mutants essentially appeared wild-type but by 12 mpf they had developed very subtle D/V curves in the pre-caudal vertebrae. They do not exhibit M/L curves at either stage.

We called the phenotype displayed by *pkd2l1^icm02^* mutants “kyphosis” to be in line with a previous publication describing these mutants (Sternberg et al., 2018). We have now added new wording in the Discussion about whether or not zebrafish can truly model kyphosis and lordosis, and we now make clear in our Results that the “mutants went on to develop very subtle kyphosis-like curves” rather than “is kyphosis”.

It is intriguing that *pkd2l1^icm02^* mutants do not exhibit any curves until much later in life than *urp1^∆P^;urp2^∆P^* and *uts2r3^b1436^* mutants. Inspired by this finding, we aged *urp1^∆P^* and *urp2^∆P^* single mutants and found that they go on to develop D/V curves by 12 mpf, with *urp1^∆P^* mutants being less severe than *urp2^∆P^* mutants (*Figure 2—figure supplement 2*). To summarize all of our findings:

**Author response table 1. sa2table1:** 

	3-months	12-months	Position of curve
*urp1^∆P^*	no curves	mild D/V curves	Mostly caudal
*urp2^∆P^*	mild D/V curves	intermediate D/V curves	Mostly caudal
*urp1^∆P^;urp2^∆P^*	severe D/V curves	severe D/V curves	Mostly caudal
*uts2r3^b1436^*	severe D/V curves	severe D/V curves	Mostly caudal
*cfap298^tm304^*	severe 3D curves	severe 3D curves	Caudal and pre-caudal
*pkd2l1^icm02^*	no curves	very mild D/V curves	Mostly pre-caudal

Phenotypes in *urp1^∆P^* and *urp2^∆P^* single mutants upon aging shows: (1) Urp1 and Urp2 are not entirely redundant in long-term spine maintenance and (2) proper Urp1/Urp2 dose is essential. We have now included these new insights.

This is an interesting question. To address it, we imaged more *uts2r3^b1436^* mutant spines and reconstructed views from dorsal aspect, including these in new *Figure 4Fi and Figure 4—figure supplement 1*. We also quantified the degree of lateral curvature (*Figure 4Fii*). The reviewer’s suggestion is correct – there are minimal side-to-side curves in *uts2r3^b1436^* mutants, highly similar to what we found for *urp1^∆P^;urp2^∆P^* double mutants.

9. One finding that deserves more discussion is the observation that urp1/urp2 double mutants have almost no side-to-side defects and all the obvious bends are in the D/V plane. Does this uncouple two systems for posture? Please consider the following paper. It shows a proprioception system that maintains normal side-to-side posture. A spinal organ of proprioception for integrated motor action feedback.Picton LD, Bertuzzi M, Pallucchi I, Fontanel P, Dahlberg E, Björnfors ER, Iacoviello F, Shearing PR, El Manira A. Neuron. 2021 Apr 7;109(7):1188-1201.e7. doi: 10.1016/j.neuron.2021.01.018. Epub 2021 Feb 11. PMID: 33577748

Thank you for pointing out this manuscript. We have now mentioned this potential “uncoupling” in our Discussion and included this citation.

Reviewer 21. Figure 3F: might be improved by making the images black and white and possibly inverted. It is not easy to clearly see the vertebrae as is.

Thanks for the suggestion, we made this change.